# Reconstructing KV Caches with Cross-Layer Fusion for Enhanced Transformers

**Hongzhan Lin[1,2,†], Zhiqi Bai[1,†], Xinmiao Zhang[1], Xiang Li[1], Siran Yang[1], Yunlong Xu[1],**
**Jiaheng Liu[3], Yongchi Zhao[1], Jiamang Wang[1], Yuchi Xu[1], Wenbo Su[1], Bo Zheng[1,✉]**
[1] Taobao & Tmall Group of Alibaba     [2] GSAI, Renmin University of China
[3] Nanjing University

## Abstract

Transformer decoders have achieved strong results across tasks, but the memory required for the KV cache becomes prohibitive at long sequence lengths. Although Cross-layer KV Cache sharing (e.g., YOCO, CLA) offers a path to mitigate KV Cache bottleneck, it typically underperforms within-layer methods like GQA. To understand the root cause, we investigate the information flow of keys and values of the top-layers. Our preliminary reveals a clear distribution: values are predominantly derived from the bottom layer, while keys draw more information from both bottom and middle layers. Building upon this, we propose FusedKV, whose top-layer KV caches are a learnable fusion of the most informative ones from the bottom and middle layers. This fusion operates directly on post-RoPE keys, preserving relative positional information without the computational cost of re-applying rotary embeddings. To further improve efficiency, we propose FusedKV-Lite, an cross-layer sharing approach, where top-layer KV caches are directly derived from the bottom-layer values and the middle-layer keys. Compared to FusedKV, FusedKV-Lite reduces I/O overhead at the cost of a slight increase in perplexity. In experiments on LLMs ranging from 332M to 4B parameters, our proposed method reduce 50% cache memory while achieving lower validation perplexity than the standard Transformer decoder, establishing it as a memory-efficient, high-performance architectural alternative. We have made our code available [1].

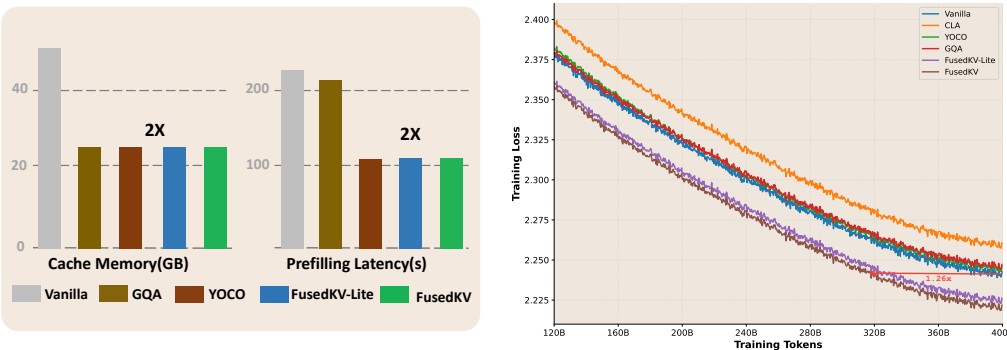

Figure 1: FusedKV and FusedKV-Lite reduce KV cache and prefilling latency by 2x (left) while also achieving superior pretraining loss on a 1.5B model compared to other methods (right).FusedKV converge around 1.26x faster than Vanilla.

## 1 Introduction

Large language models (LLMs) based on the Transformer (Vaswani et al., 2017) decoder have achieved unprecedented success across a wide range of tasks and applications. However, practi-

[†] Equal contribution    ✉ Corresponding author
[1]https://github.com/LivingFutureLab/FusedKV

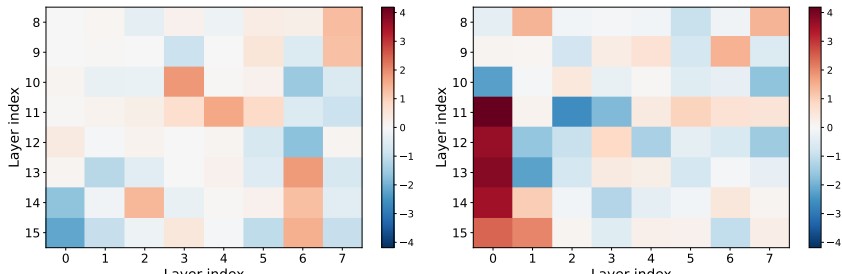

Figure 2: Fusion weight for reconstructing key (left) and value (right) caches in the top 8 layers of a 16-layer model. The figure reveals a clear asymmetry in key-value caches.

cal deployment for real-world applications requires LLM inference to achieve low latency and high throughput (Hu et al., 2024; Ge et al., 2024; Dao et al., 2022). This becomes particularly challenging with increasingly long contexts, primarily due to the linear memory overhead of the key-value (KV) caches in autoregressive generation (Liu et al., 2024b; Adnan et al., 2024; Li et al., 2024).

To mitigate this, various methods have been proposed to reduce KV caches. Within the layer, Group-Query Attention (GQA [2]) (Ainslie et al., 2023) and Multi-Query Attention (MQA) (Shazeer, 2019) reduce redundancy by sharing one head's key and value across multiple query heads. MLA (Liu et al., 2024a) uses low-rank joint compression for key-value caches. Across the layer, CLA (Brandon et al., 2024) shares the KV cache between adjacent layers, while YOCO [3] (Sun et al., 2024) reuses the intermediate layer cache for the top-half layers. While cross-layer sharing techniques also reduce memory overhead, they consistently underperform within-layer methods, which motivates the need for a more effective cross-layer sharing strategy.

In this paper, we reveal a previously overlooked asymmetric key-value sharing principle: for top-half layers where values are most effectively reconstructed by the bottom layer, whereas keys draw their more critical information from the bottom and middle layer. Motivated by this, we propose FusedKV, which reconstructs the top-half layer caches by performing a dimension-wise, weighted fusion of caches from two highly informative source layers: the bottom and middle layer. This fusion can be applied directly to post-RoPE keys, a design that preserves relative positional information without the computational overhead of RoPE recomputation. FusedKV halves the cache size while achieving significantly lower perplexity compared to the full-cached model, as shown in Figure 1. We further provide a more effecient version, FusedKV-Lite, where top-half layers' keys are directly reused from the middle layer and values from the bottom layer. Compared to FusedKV, FusedKV-Lite decreases I/O overhead by one-third with only a slight degradation in performance.

Our contributions are threefold: (1) We identify a key-value asymmetry in cache reconstruction. (2) Based on this insight, we propose FusedKV, a memory-efficient architecture, and its more efficient version FusedKV-Lite. Extensive experiments demonstrate that our methods can reduce KV cache memory by 50% while achieving lower perplexity than a standard Transformer decoder. (3) We provide an efficient Triton implementation of FusedKV and a systematic evaluation of KV cache reduction methods, including performance efficiency comparisons and integration of our approach with existing architectures.

## 2 METHOD

In this section, we first define the overall framework for cross-layer KV cache sharing. We then introduce two specific methods, FusedKV-Lite and FusedKV, and demonstrate FusedKV's compatibility with RoPE. Finally, we provide a complexity analysis of different KV sharing methods.

---

[2]Unless otherwise noted, GQA mentioned in this paper uses a configuration where two query heads share a single key-value head.

[3]Note that in the original YOCO paper, the cache-storing layers use efficient attention mechanisms such as Sliding Window Attention(SWA). In this paper, unless otherwise stated, we use full attention.

## 2.1 OVERALL FRAMEWORK DEFINITION

We partition the set of $L$ decoder layers, $\mathcal{L} = \{1, \ldots, L\}$, into two disjoint subsets: **Storage Layers** ($\mathcal{L}_S$), for which the KV caches are explicitly stored in memory during generation and **Reconstruction Layers** ($\mathcal{L}_R$), whose KV caches are not stored but are recomputed on-demand via a reconstruction function. For any reconstruction layer $i \in \mathcal{L}_R$, its Key, $\boldsymbol{K}^i \in \mathbb{R}^{s \times d}$, and Value, $\boldsymbol{V}^i \in \mathbb{R}^{s \times d}$, are generated by a parameterized **reconstruction function** $\mathcal{F}_i$ using the KV caches from a subset of the storage layers. The formula is as follows:

$$(\boldsymbol{K}^i, \boldsymbol{V}^i) = \mathcal{F}_i \left( \{(\boldsymbol{K}^j, \boldsymbol{V}^j) | j \in \Phi(i)\}; \theta_i \right) \tag{1}$$

where $\Phi(i)$ is a **Source Layer Mapping Function** that specifies a set of source storage layer indices for reconstruction layer $i$. The trainable parameters $\theta_i$ associated with the $i$-th reconstruction layer could be data-dependent or data-independent.

## 2.2 RECONSTRUCTION FUNCTION $\mathcal{F}$

The architecture of the reconstruction function $\mathcal{F}$ determines the complexity and effectiveness of the fusion process. We explore two primary categories for the reconstruction function $\mathcal{F}$, differing in their complexity and parameterization.

**Direct Cache Reuse.** The most straightforward approach for reconstruction is to directly reuse the KV cache from the source layers without any transformation. In this case, The reconstruction can be formulated as:

$$(\boldsymbol{K}^i, \boldsymbol{V}^i) = (\boldsymbol{K}^{\phi(i)}, \boldsymbol{V}^{\phi(i)}), \quad \text{where } \phi(i) \in \mathcal{L}_S. \tag{2}$$

the reconstruction function $\mathcal{F}_i$ acts as a selector, the source layer mapping $\Phi(i)$ typically contains a single index, i.e., $|\Phi(i)| = 1$. Recent methods such as YOCO (Sun et al., 2024) and CLA (Brandon et al., 2024) adopt this approach.

- **YOCO** partitions layers into two contiguous blocks: a storage block ($\mathcal{L}_S = \{1, \ldots, L/2\}$) and a reconstruction block ($\mathcal{L}_R = \{L/2 + 1, \ldots, L\}$). All reconstruction layers access the KV cache from the final storage layer, defined by the constant mapping $\phi(i) = L/2$.
- **CLA** employs an interleaved partitioning scheme, designating odd-numbered layers for storage ($\mathcal{L}_S = \{1, 3, \ldots, L - 1\}$) and even-numbered layers for reconstruction ($\mathcal{L}_R = \{2, 4, \ldots, L\}$). Each reconstruction layer $i$ accesses the KV cache from its immediately preceding storage layer, with the mapping $\phi(i) = i - 1$.

**Weighted Fusion of Caches.** Direct cache reuse may lead to representation collapse in shared layers, potentially limiting their layer-specific contributions in forward flowing. A more expressive method involves computing the reconstructed KV cache as a weighted linear combination of the caches from multiple source layers. The general form is:

$$\boldsymbol{K}^i = \sum_{j \in \Phi(i)} \boldsymbol{a}_{ij} \odot \boldsymbol{K}^j, \quad \boldsymbol{V}^i = \sum_{j \in \Phi(i)} \boldsymbol{b}_{ij} \odot \boldsymbol{V}^j. \tag{3}$$

Where the learnable weight $\boldsymbol{a}_{ij}$ and $\boldsymbol{b}_{ij}$ can be scalars ($\mathbb{R}$), vectors ($\mathbb{R}^d$), or matrices ($\mathbb{R}^{s \times d}$), and are broadcast to match the shape of the Key and Value caches for the Hadamard product $\odot$. In this way, we reconstruct the target cache by applying a feature-wise gating mechanism, selectively re-weighting features from each source layer before their aggregation.

## 2.3 FUSEDKV AND FUSEDKV-LITE

**Preliminary experiments.** We conduct a dense fusion experiment on a 1B-parameter, 16-layer LLM. The top 8 layers reconstructs its cache from a learnable *scalar* fusion of all the bottom caches. As shown in Figure 10 of the Appendix, dense fusion attains a lower training loss than the vanilla, indicating that top-layer KV caches can be effectively reconstructed from earlier layers. As depicted in Figure 2, we also find a clear asymmetry between keys and values: value-cache fusion is dominated by layers 0–1 for reconstruction layers 10–15, while key-cache weights are more diffuse but concentrate on source layers 6–7. These results indicate that the bottom and middle layers are more informative for reconstructing top-layer caches.

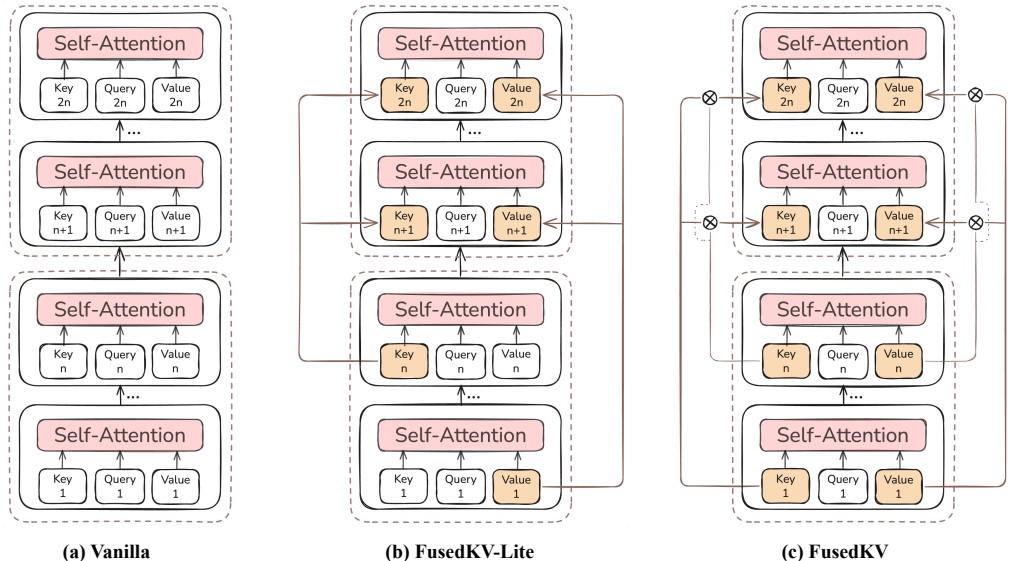

Figure 3: Illustration of KV cache strategies. (a) Vanilla: The standard method with a unique KV cache for each layer. (b) FusedKV-Lite: For layers $i > n$, the Key cache is reused from layer $n$, and the Value cache from layer 1. (c) FusedKV: For layers $i > n$, the caches are a learnable weighted fusion (denoted by $\otimes$) of the caches from layer 1 and layer $n$.

**FusedKV.** Motivated by the observation in dense fusion experiment, we propose a simplified yet effective strategy named FusedKV. As illustrated in Figure 3(c), FusedKV reconstructs the cache by computing a learnable weighted fusion of caches from two highly informative source layers: the bottom (layer 1) and the middle (layer $n$). The process is formulated as:

$$\boldsymbol{K}^i = \boldsymbol{a}_{i,1} \odot \boldsymbol{K}^1 + \boldsymbol{a}_{i,n} \odot \boldsymbol{K}^n, \quad i > n. \tag{4}$$

$$\boldsymbol{V}^i = \boldsymbol{b}_{i,1} \odot \boldsymbol{V}^1 + \boldsymbol{b}_{i,n} \odot \boldsymbol{V}^n, \quad i > n. \tag{5}$$

FusedKV allows each reconstruction layer to synthesize its cache from both the low-level, foundational features of the initial layer and the more abstract, contextual representations from the final source layer. This approach strikes an effective balance between representational power and the memory traffic costs associated with cache fusion.

**FusedKV-Lite.** To further improve efficiency, we reconstruct the keys by directly reusing the cache from the last source layer $n$, and the values from the bottom layer, as illustrated in Figure 3(b). The reconstruction is formulated as:

$$\boldsymbol{K}^i = \boldsymbol{K}^n, \quad \boldsymbol{V}^i = \boldsymbol{V}^1, \quad i > n. \tag{6}$$

By reusing only a single source key and value cache, FusedKV-Lite avoids the additional I/O overhead from fusion, thereby maintaining efficiency on par with the vanilla model, making it highly efficient for I/O-bound inference scenarios.

## 2.4 RoPE Compatibility

In this section, we establish that RoPE is compatible with learnable weights when the weights are symmetric, and we further show that fusing post-RoPE key caches preserves relative positional encoding.

**Weight vectors must be 2D-diagonal.** To ensure that learnable weights do not disrupt the relative positional encoding property of RoPE (Su et al., 2024), we analyze the attention score. For a query at position $m$ and a key at position $n$, the attention score $A_{n,j}$ in the $j$-th 2D subspace, after incorporating a learnable weight vector $\boldsymbol{w}_j = [w_{2j}, w_{2j+1}]^T$, can be decomposed as follows (see

Appendix A.4 for the full derivation):

$$A_j = \frac{w_{2j} + w_{2j+1}}{2} \Big[ (q_{m,2j}k_{n,2j} + q_{m,2j+1}k_{n,2j+1}) \cos((m-n)\theta_j) \tag{7}$$

$$+ (q_{m,2j}k_{n,2j+1} - q_{m,2j+1}k_{n,2j}) \sin((m-n)\theta_j) \Big]$$

$$+ \frac{w_{2j} - w_{2j+1}}{2} \Big[ (q_{m,2j}k_{n,2j} - q_{m,2j+1}k_{n,2j+1}) \cos((m+n)\theta_j)$$

$$- (q_{m,2j}k_{n,2j+1} + q_{m,2j+1}k_{n,2j}) \sin((m+n)\theta_j) \Big]$$

Equation 7 reveals that when $w_{2j} \neq w_{2j+1}$, the attention score becomes a mixture of a relative position term (dependent on $m - n$) and an absolute position term (dependent on $m + n$). To preserve RoPE's pure relative position dependency, we must enforce identity within each 2D weight pair, i.e., $w_{2j} = w_{2j+1}$.

**Weighted Fusion Preserves RoPE.** With the symmetry constraint established, we show that a weighted fusion of multiple RoPE-transformed key vectors maintains relative positional encoding. Let $\tilde{k}_s = \sum_{i=1}^{N}(w_n^i \odot \tilde{k}_n^i)$ be a fused key vector from $N$ different storage layers, where each weight vector $w_n^i$ is symmetric ($w_{n,2j}^i = w_{n,2j+1}^i$). The attention score with this fused key is:

$$\tilde{q}_m^T \tilde{k}_s = \tilde{q}_m^T \left( \sum_{i=1}^{N}(w_n^i \odot \tilde{k}_n^i) \right) = \sum_{i=1}^{N} \tilde{q}_m^T(w_n^i \odot \tilde{k}_n^i) \tag{8}$$

As established by Equation 7 with the symmetry constraint, each term $\tilde{q}_m^T(w_n^i \odot \tilde{k}_n^i)$ in the sum is a function of only the content vectors and the relative position $m - n$. Consequently, their linear combination, $\tilde{q}_m^T \tilde{k}_s$, also upholds this property. This compatibility is practically significant. It allows storage layers to retain their original post-RoPE KV caches, avoiding re-computing RoPE at inference time.

Table 1: Complexity analysis. $L$ denotes the number of model layers, $S$ denotes the sequence length, $D$ denotes the head dimension, and $H_q$ and $H_{kv}$ represents the number of query heads and key-value heads, respectively.

| Method | Prefilling FLOPs | Decoding FLOPs | Cache Mem. | Cache I/O. |
|---|---|---|---|---|
| MHA/GQA | $LSH_qD(4S + 4H_qD + 4H_{kv}D)$ | $LH_qD(4S + 4H_qD + 4H_{kv}D)$ | $2LSH_{kv}D$ | $2LSH_{kv}D$ |
| YOCO | $LSH_qD(2S + 2H_qD + 2H_{kv}D + 2) + 2L(HqD)^2$ | $LH_qD(4S + 4H_qD + 2H_{kv}D)$ | $LSH_{kv}D$ | $2LSH_{kv}D$ |
| FusedKV-Lite | $LSH_qD(2S + 2H_qD + 2H_{kv}D + 2) + 2L(HqD)^2$ | $LH_qD(4S + 4H_qD + 2H_{kv}D)$ | $LSH_{kv}D$ | $2LSH_{kv}D$ |
| FusedKV | $LSH_qD(2S + 2H_qD + 2H_{kv}D + 2 + 3\frac{H_{kv}}{H_q}) + 2L(HqD)^2$ | $LH_qD(4S + 4H_qD + 2H_{kv}D + 3S\frac{H_{kv}}{H_q})$ | $LSH_{kv}D$ | $3LSH_{kv}D$ |

## 2.5 INFERENCE

We conduct a complexity analysis of different attention mechanisms in Table 1. The comparison focuses on the computational cost in prefilling and decoding phase, KV cache memory footprint, and KV cache I/O volume. These calculations are confined to the attention component, excluding the feed-forward networks. We find that both FusedKV-Lite and FusedKV reduce prefilling FLOPs and cache memory compared to the vanilla model. Due to its fusion computation, FusedKV incurs a slightly higher cache I/O volume compared to the other methods.

We implement a Triton-based attention kernel for proposed FusedKV operator, and benchmark the performance of FusedKV and FusedKV-Lite from three aspects: (i) attention throughput, which captures the speed of token generation by the attention kernel, (ii) end-to-end prefill performance, which is quantified by Time to First Token (TTFT), and (iii) end-to-end decoding throughput, which is expressed by Time Per Output Token (TPOT).

**Attention Throughput.** As shown in Figure 4 (left), the throughput of FusedKV kernel is 28.4% lower on average than MHA due to an extra cache I/O. In contrast, FusedKV-Lite, which maintains a comparable cache I/O, achieves identical throughput to MHA.

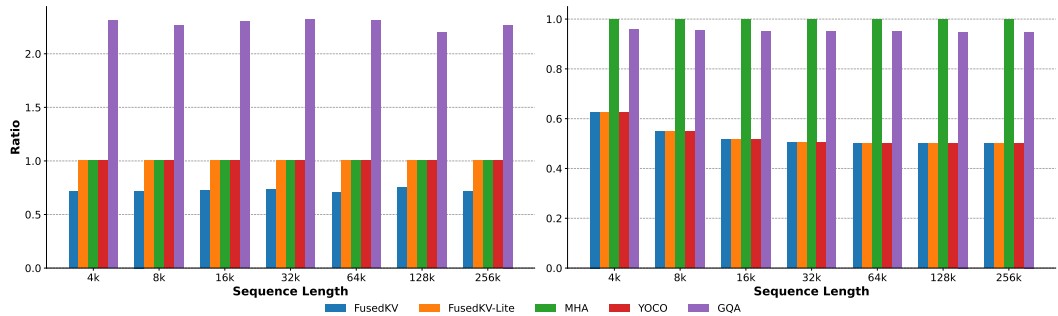

Figure 4: Left: The attention throughput of different kernels (The higher is better). Right: Time to First Token (TTFT) of different models, showing the end-to-end prefilling performance (The lower is better). All methods are normalized by the vanilla (MHA) baseline.

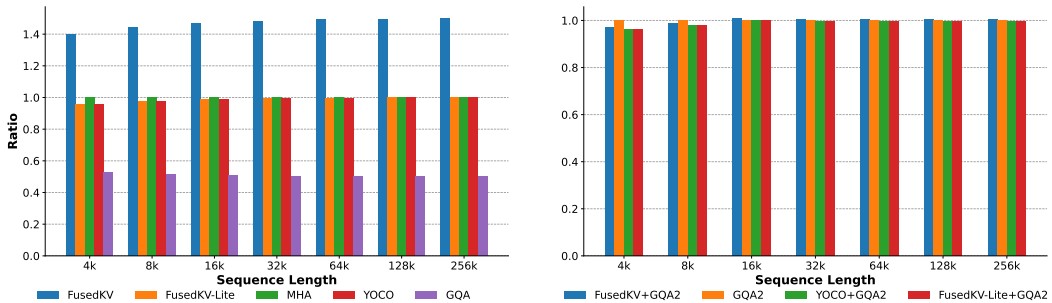

Figure 5: Time Per Output Token (TPOT) performance ratios. The left panel displays the memory-bound scenario, and the right panel displays the compute-bound scenario.

**Time to First Token (TTFT).** As shown in Figure 4 (right), both FusedKV and FusedKV-Lite exhibit a clear advantage over vanilla implementation on prefilling latency. Specifically, for input sequence length of 8k and beyond, the TTFT is reduced by approximately 50% relative to vanilla. This improvement originates from FusedKV and FusedKV-Lite layers, which reuse the KV cache from preceding layers and skip the cache prefilling in current layers, thus halving the overall prefilling latency for the end-to-end model.

**Time Per Output Token (TPOT).** We further examine the decoding speed under both memory-bound and compute-bound scenarios. As illustrated in Figure 5 (left), in the memory-bound setting, the additional cache I/O overhead in the FusedKV layers leads to an approximately $1.5\times$ increase in TPOT compared to the vanilla implementation. In compute-bound settings, where the baseline uses GQA with $H_q \gg H_{kv}$, this cache I/O overhead is effectively hidden by the computational workloads. For evaluation, we adopt GQA with 128 query heads and 2 key–value heads as the baseline. The computational overhead introduced by cache fusion accounts for only $\frac{3 \times H_{kv}}{4 \times H_q} = \frac{3}{256}$ of that required by attention operation, which is practically negligible. Consequently, TPOT of FusedKV remains comparable to that of the baseline. In contrast, FusedKV-Lite maintains a cache I/O nearly identical to that of the vanilla implementation. Therefore, its TPOT is similar to the baseline under both memory-bound and compute-bound scenarios.

## 3 EXPERIMENTS

### 3.1 GENERAL SETUP

**Architecture and Training Details** We introduce three dense language models with 332M, 650M and 1.5B parameters, both following the Qwen3 architecture (Team, 2025). More configuration are detailed in Table 6. All three models share the same configuration: a vocabulary of 128,000 tokens, a context length of 8,192, and 16 attention heads. They were trained on the FineWeb-Edu dataset (Lozhkov et al., 2024). We use the AdamW optimizer (Loshchilov & Hutter, 2017) with $\beta_1 = 0.9$, $\beta_2 = 0.95$. The learning rate followed a cosine schedule, warming up for 2,000 steps to

Table 2: Evaluation of various KV cache reduction methods on downstream tasks, highlighting the competitive performance of FusedKV and FusedKV-Lite across 332M, 650M, and 1.5B model sizes.

| | Model | Cache Mem. ↓ | Valid Loss ↓ | Wiki Text ↓ | MNLI | SCIQ | LAMB-Acc | Hella Swag | ARC-E | ARC-C | MMLU | Avg Acc↑ |
|---|---|---|---|---|---|---|---|---|---|---|---|---|
| **332M params.** | Vanilla | 1 | 2.651 | 22.85 | 34.64 | 84.20 | 24.49 | 42.12 | 59.26 | 29.69 | 28.93 | 43.33 |
| | CLA | 1/2 | 2.676 | 24.62 | 34.80 | 76.70 | 13.51 | 40.75 | 54.67 | 27.65 | 28.56 | 39.52 |
| | YOCO | | 2.663 | 23.31 | 33.64 | **80.60** | 26.61 | 41.74 | 59.93 | 30.29 | 28.77 | 43.08 |
| | GQA | | 2.662 | 23.29 | **34.99** | 80.50 | 25.52 | 41.72 | 60.23 | 31.40 | **29.45** | 43.40 |
| | FusedKV-Lite | | **2.639** | 22.78 | 34.22 | 79.60 | 26.39 | 42.07 | 58.67 | **32.00** | 29.38 | 43.19 |
| | FusedKV | | 2.642 | **22.35** | 33.62 | 80.30 | **28.10** | **42.74** | **60.98** | 30.12 | 29.26 | **43.59** |
| **650M params.** | Vanilla | 1 | 2.483 | 18.47 | 34.39 | 88.10 | 31.67 | 49.13 | 65.36 | 36.26 | 31.62 | 48.08 |
| | CLA | 1/2 | 2.511 | 19.60 | 34.25 | 86.10 | 27.94 | 47.48 | 65.91 | 32.94 | 30.81 | 46.49 |
| | YOCO | | 2.498 | 19.21 | **34.96** | 86.70 | 30.74 | 48.65 | 64.14 | 34.39 | 30.66 | 47.18 |
| | GQA | | 2.497 | 19.05 | 33.11 | 86.30 | 30.56 | 48.62 | 65.36 | 34.04 | 31.56 | 47.08 |
| | FusedKV-Lite | | **2.473** | 18.55 | 34.24 | **86.90** | **32.95** | 49.53 | **66.58** | **37.46** | **31.88** | **48.51** |
| | FusedKV | | 2.474 | **18.09** | 33.33 | 86.60 | 31.03 | **49.68** | 65.61 | 34.39 | 31.54 | 47.45 |
| **1.5B params.** | Vanilla | 1 | 2.241 | 13.67 | 35.69 | 94.70 | 40.54 | 60.67 | 72.55 | 42.58 | 35.12 | 54.55 |
| | CLA | 1/2 | 2.258 | 14.19 | 35.00 | 92.60 | 39.61 | 59.04 | 73.99 | 42.32 | 34.83 | 53.91 |
| | YOCO | | 2.244 | 13.65 | 35.30 | 91.70 | 40.81 | 60.09 | 73.19 | 42.92 | 35.35 | 54.19 |
| | GQA | | 2.245 | 13.74 | 34.63 | 92.40 | 41.20 | 60.16 | 73.91 | **44.28** | 35.49 | 54.58 |
| | FusedKV-Lite | | 2.225 | 13.45 | 36.43 | **93.70** | 41.61 | 60.77 | **74.79** | 43.52 | **36.29** | 55.30 |
| | FusedKV | | **2.221** | **13.33** | 37.43 | 93.50 | **43.33** | **61.88** | 74.20 | 44.20 | 36.18 | **55.82** |

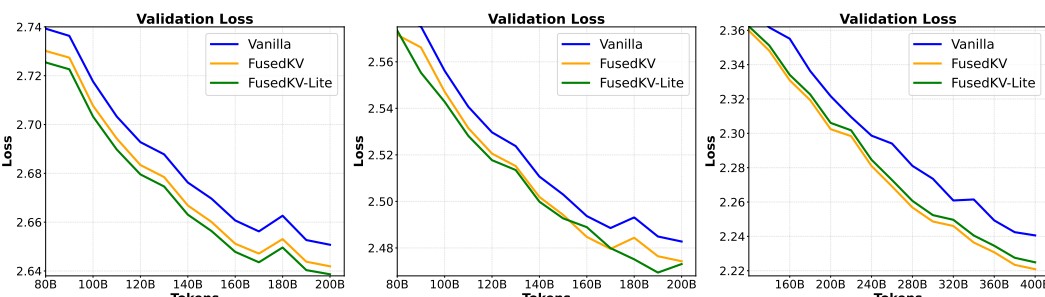

Figure 6: Validation loss curves of 332M (left), 650M (center), and 1.5B (right) dense models, where FusedKV and FusedKV-Lite consistently achieves lower validation loss than the Vanilla.

a peak of $3 \times 10^{-4}$ and decaying to a minimum of $3 \times 10^{-5}$. A more detailed configuration can be found in Table 4.

**Evaluation** For model evaluation and comparison, we assess perplexity on a 500M-token validation split randomly sampled from FineWeb-Edu. We also evaluate perplexity on WikiText (Merity et al., 2016). We also conduct five-shot evaluation across diverse language tasks, including MNLI (Williams et al., 2018), SCIQ (Welbl et al., 2017), LAMBADA (LAMB-Acc) (Paperno et al., 2016), Hellaswag (Zellers et al., 2019), ARC-Easy (Clark et al., 2018), ARC-Challenge (Clark et al., 2018) and MMLU (Hendrycks et al., 2021). All evaluations are based on the LM Evaluation Harness (Gao et al.).

## 3.2 MAIN RESULTS

As presented in Table 2, our proposed methods, FusedKV-Lite and FusedKV, demonstrate strong performance across three model scales. Compared to the full-cache Vanilla baseline and other cache-saving methods, our approaches consistently achieve competitive or superior results on both language modeling perplexity and downstream task accuracy.

The effectiveness of our methods is particularly pronounced at the 1.5B scale. FusedKV not only achieves the lowest validation loss (2.221) and WikiText perplexity (13.33) but also, alongside FusedKV-Lite, secures the highest average downstream accuracies (55.82 and 55.30, respectively). Notably, these results substantially outperform the full-cache vanilla baseline (54.55) and include top performance on challenging tasks such as ARC-E, MMLU, and HellaSwag.

This trend of strong performance remains consistent across smaller model sizes. At the 332M and 650M scales, FusedKV-Lite and FusedKV respectively lead all methods in average downstream ac-

Table 3: FusedKV outperforms the vanilla on 4B-parameter LLMs in perplexity and downstream tasks.

| Model | Valid Loss ↓ | Wiki Text ↓ | MNLI | SCIQ | LAMB- Acc | Hella Swag | ARC-E | ARC-C | MMLU | Avg Acc↑ |
|-------|------|------|------|------|------|------|------|------|------|------|
| Vanilla | 2.002 | 9.18 | 37.27 | **96.00** | 49.60 | 68.92 | 76.43 | 46.59 | 38.71 | 59.07 |
| FusedKV | **1.978** | **8.94** | **38.52** | 95.20 | **50.18** | **69.94** | **77.78** | **48.63** | **39.83** | **60.01** |

curacy, though we note their performance on challenging tasks like MMLU and ARC-C remains close to the chance level. Crucially, when compared to methods with equivalent 50% cache savings like YOCO and GQA, our approaches consistently yield superior accuracy. These comprehensive results, further illustrated by the validation loss curves in Figure 6, highlight the ability of our methods to enhance model performance and in-context learning capabilities. Furthermore, we present results for long-context scenarios in Appendix A.7.

## 3.3 SCALING LAW EXPERIMENTS

We compare the loss scaling behavior of the vanilla and FusedKV across increasing model parameters ranging from 332M to 4B. The training configurations used in scaling experiments are detailed in Table 4 of the Appendix, with model sizes and token counts detailed in Table 6. Specifically, the 332M, 650M, 1.5B and 4B models are trained on 200B, 200B, 400B, and 800B tokens, respectively. As illustrated in Figure 7, FusedKV achieves better scaling efficiency, with its loss decreasing more notably as the model capacity grows. For 4B-parameter LLMs, FusedKV exhibits better downstream performance than vanilla, as depicted in Table 3. This demonstrates its ability to retain performance at larger scales, making it particularly promising for large-parameter models.

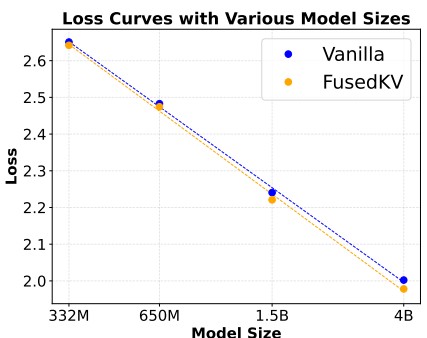

Figure 7: Scaling law curves of FusedKV and the vanilla.

## 3.4 EXTENSIBILITY AND COMPATIBILITY

We extensively validate the compatibility of FusedKV and FusedKV-Lite across a diverse spectrum of efficiency-oriented designs, including Multi-Head Latent Attention (MLA) (Liu et al., 2024a), Grouped-Query Attention (GQA) (Ainslie et al., 2023), Mixture-of-Experts (MoE) (Shazeer et al., 2017), and hybrid structures like Sliding Window Attention (SWA) (Jiang et al., 2023). Empirical results demonstrate that our cross-layer fusion mechanism is largely orthogonal to these architectural optimizations, often yielding synergistic benefits in both inference and memory footprint with negligible performance degradation. Full experimental details, configurations, and results are provided in Appendix A.6.

## 3.5 ABLATION STUDY

In this section, we investigate two key aspects of our method: the directionality of the asymmetric KV assignment and the impact of parameterizing the reconstruction with learnable weights. We conduct a more detailed ablation study in Appendix A.8.

**Directionality of KV asymmetry.** We first investigate the impact of the asymmetric assignment strategy. As shown in Figure 8, reversed assignment (FusedKV-Lite-Rev, where $K^i = K^1, V^i = V^8, i > 8$) degrades performance substantially compared with our proposed FusedKV-Lite ($K^i = K^8, V^i = V^1, i > 8$). Compared with FusedKV-Lite, FusedKV-Lite-Rev exhibits markedly higher validation loss and commensurately lower downstream accuracy. This gap demonstrates that when directly reusing a single-source KV cache, later-layer Keys and early-layer Values are more informative for reproducing the original behavior.

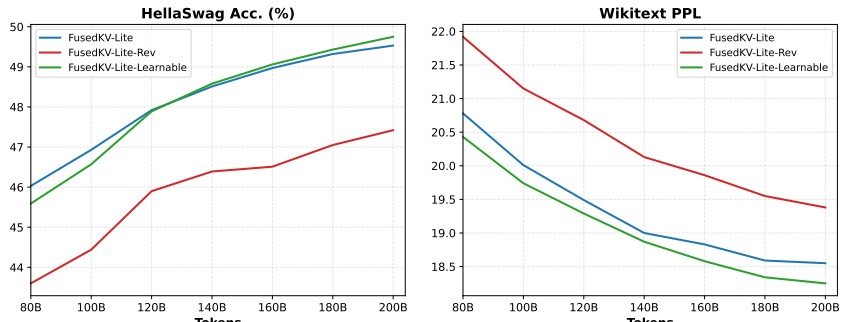

Figure 8: Ablation study of KV asymmetry and learnable weights on 650M dense models with 200B tokens. We report the perplexity on Wikitext and the five-shot accuracy on Hellaswag for three key variants: FusedKV-Lite, FusedKV-Lite-Rev, and FusedKV-Lite-Learnable.

**Impact of learnable weights.** We also study the impact of parameterizing the reconstruction with learnable weights. The variant, FusedKV-Lite-Learnable, uses learnable vectors to perform channel-wise re-weighting: $\boldsymbol{K}^i = \boldsymbol{a}_{i8} \odot \boldsymbol{K}^8$ and $\boldsymbol{V}^i = \boldsymbol{b}_{i1} \odot \boldsymbol{V}^1$ for layer indices $i > 8$. The results in Figure 8 show that FusedKV-Lite-Learnable (green line) improves upon the fixed-weight FusedKV-Lite baseline (blue line). It achieves lower perplexity on both Wikitext and LAMBADA. FusedKV-Lite-Learnable also outperforms FusedKV-Lite on Hellaswag in the later stages of training. This demonstrates that allowing the model to adaptively re-weight source Key and Value channels increases expressiveness and yields downstream gains.

## 4 GRADIENT VISUALIZATION

We visualize the L2 norm of gradients for the query, key, and value projection matrices at different network depths. As depicted in Figure 9, we track these gradient norms throughout the training of a 650M-parameter model. The results reveal a distinct pattern: **FusedKV** and **FusedKV-Lite** consistently exhibit a markedly larger gradient L2 norm compared to baselines, particularly within the shallower layers of the network (e.g., Layer 1 and Layer 5). A larger gradient magnitude implies more substantial parameter updates during backpropagation. Therefore, the pronounced gradients in the shallow layers for FusedKV and FusedKV-Lite suggest that these layers are learning more effectively and rapidly. By facilitating stronger gradient signals to the initial layers, our learnable fusion mechanism enables the model to more quickly refine its foundational representations. This suggests that the primary benefit of our fusion mechanism on gradient flow is to accelerate learning in the crucial early layers, which form the bedrock for the hierarchical features learned by the deeper parts of the network.

## 5 RELATED WORK

### 5.1 CROSS-LAYER KV CACHE

KV caching is a major memory bottleneck in LLM inference (Wu et al.; Liao & Vargas, 2024). Memory footprint can be reduced by limiting KV state retention to a subset of layers. YOCO (Sun et al., 2024) reuses a global KV cache from an intermediate layer for the model's latter half, roughly halving memory; CLA (Brandon et al., 2024) shares caches across adjacent layers. More aggressive cross-layer reuse includes Mixture of Recurrent (Bae et al., 2025), which stacks recurrent blocks (each comprising multiple transformer layers) so later blocks reuse the first block's KV. SVFormer (Zhou et al., 2025), which does not share the key but reuses the bottom layer's values; and LCKV (Wu & Tu, 2024), which caches only the last layer's KV for all earlier layers, with safeguards against cyclic dependencies. However, most approaches often overlook the distinct roles of keys and values: keys drive relevance scoring, while values supply contextual content. Treating KV as a single unit for caching neglects this functional disparity. Moreover, sharing KV states indiscriminately across layers risks undermining each layer's specific role in hierarchical processing, potentially introducing irrelevant context and hindering its unique function. These critical aspects remain underexplored.

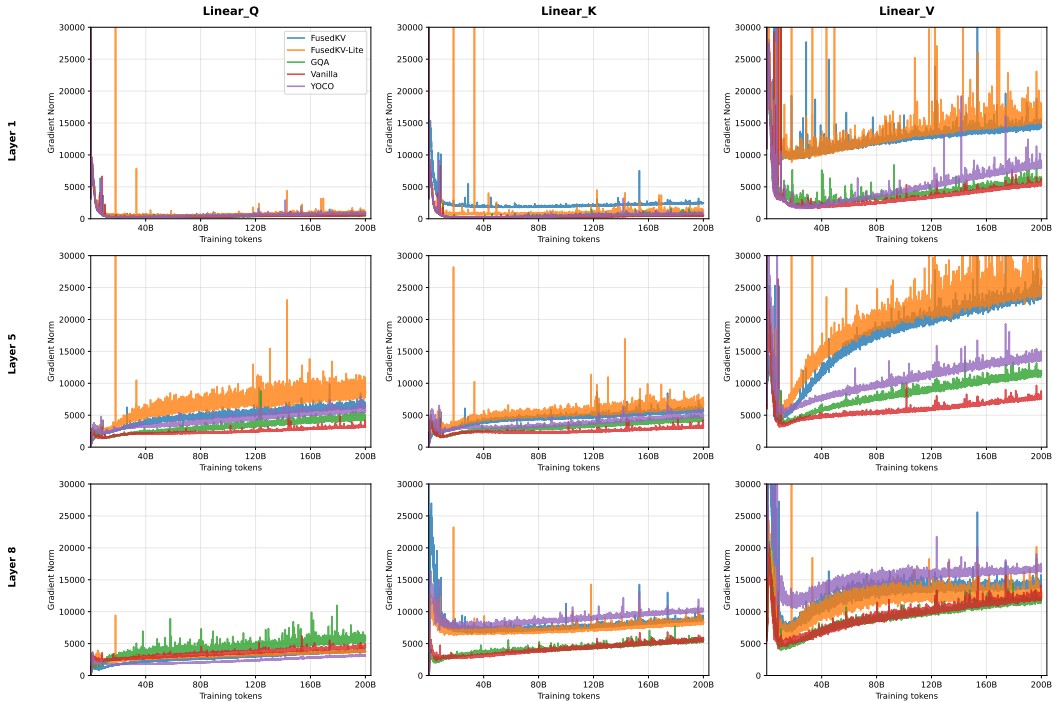

Figure 9: Comparison of gradient L2 norms shows that FusedKV and FusedKV-Lite maintain significantly stronger gradient flow in shallow layers compared to baselines. This suggests a healthier gradient flow that accelerates the convergence of early layers.

## 5.2 EARLY-FEATURE FUSION

Another line of research leverages early-layer feature fusion to stabilize gradients and enrich global features( (He et al., 2016; Huang et al., 2017; Pagliardini et al., 2024; Abdullaev & Nguyen, 2025)). ResNet (He et al., 2016) introduces residual connections that add outputs from previous layers to the current layer to stabilize training. Subsequent approaches further enhance the information flow within this framework. DenseNet (Huang et al., 2017) concatenates the outputs of all previous layers, while Denseformer (Pagliardini et al., 2024) and MambaMixer (Behrouz et al., 2024) incorporate learnable weighted coefficients into the residual paths, making the residual representations more expressive. Furthermore, reusing features from earlier layers can help alleviate the over-smoothing problem (Wang et al., 2022; Shi et al., 2022), where representations converge and lose diversity as the depth of the network increases, leading to degraded performance. To mitigate this issue, Neu-TRENO (Nguyen et al., 2023) introduces a skip connection that injects a fraction of the bottom-layer output into every subsequent self-attention layer to mitigate over-smoothing. Similarly, (Abdullaev & Nguyen, 2025) enriches the transformer's representational capacity by reusing value-residual information through a twice-attention mechanism.

## 6 CONCLUSION

In this work, we identified a key-value asymmetry principle, revealing that keys and values in upper layers are best reconstructed from different source layers. Based on this insight, we proposed FusedKV, an efficient architecture that reconstructs the top-half layer caches via a weighted fusion of caches from the bottom and middle layers. We also introduced FusedKV-Lite, a more I/O-efficient variant that uses direct asymmetric sharing. Our extensive experiments demonstrate that FusedKV reduces the KV cache by 50% while remarkably achieving lower perplexity than the full-cached baseline. By offering substantial memory savings without a trade-off in performance, FusedKV and FusedKV-Lite establish a effective, integrable paradigm for cross-layer cache sharing, paving the way for more efficient deployment of powerful long-context language models.

## 7 ETHICS STATEMENT

The authors have read and adhered to the ICLR Code of Ethics. This work utilizes publicly available datasets and does not involve any sensitive personal information or experiments with human subjects. We have considered the potential societal impacts of our research and believe the work does not raise significant ethical concerns. We encourage the responsible use of our methods and findings.

## 8 REPRODUCIBILITY STATEMENT

To ensure reproducibility, the source code for our experiments has been submitted as supplementary material. It is accessible via an anonymous link and includes scripts to replicate our main results.

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

# A APPENDIX

## A.1 LLM USAGE STATEMENT

In the preparation of this manuscript, we employ Large Language Models for the purpose of language editing and refinement. The use of the LLM was strictly confined to improving grammatical accuracy, sentence structure, and overall readability of the author-written text. The authors have carefully reviewed and revised all LLM-generated suggestions and assume full responsibility for the entire content and integrity of this paper.

## A.2 ARCHITECTURES OF DIFFERENT MODELS

we adopt a decoder-only transformer architecture akin to Qwen3 (Team, 2025), with model sizes ranging from 332M to 4B parameters. The specific architecture of models is summarized in Table 6. The hyperparameters for Pretraining is shown in Table 4.

Table 4: Hyperparameters for Pretraining.

|  | Dense Model |
| --- | --- |
| Optimizer | AdamW |
| Learning Rate (LR) | 3e-4 |
| Minimum LR | 3e-5 |
| LR Schedule | cosine |
| Weight Decay | 0.1 |
| $\beta_1$ | 0.9 |
| $\beta_2$ | 0.95 |
| Gradient Clipping | 1 |
| Batch Size | 512 |
| Warmup steps | 2,000 |
| Init std | 0.02 |
| RoPE $\theta$ | 10000 |
| Activation | SwiGLU |

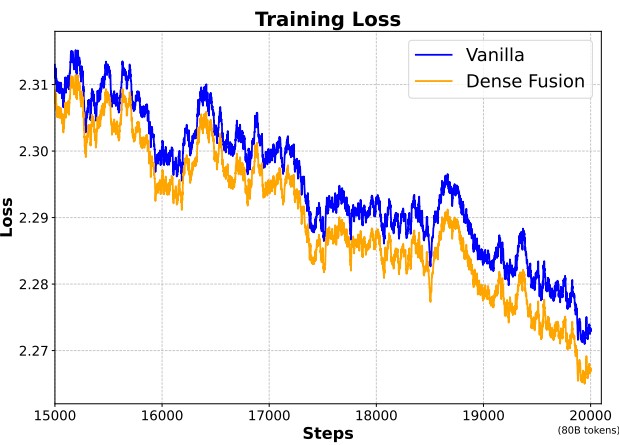

Figure 10: Training loss of a 16 layer 1B-paramter model pretarined on 80B tokens.

## A.3 TRAINING AND INITIALIZATION STRATEGIES FOR FUSEDKV

In this section, we provide a detailed description of the different training and initialization strategies explored for FusedKV. These strategies vary in their architectural implementation and weight initialization schemes, offering different trade-offs between layer-wise dependency and training simplicity.

Table 5: Performance comparison of three initialization methods with 650M-parameter model trained with 200B tokens.

| method | Valid Loss $\downarrow$ | Wiki Text $\downarrow$ | MNLI | SCIQ | LAMB-Acc | Hella Swag | ARC-E | ARC-C | MMLU | Avg Acc$\uparrow$ |
|---|---|---|---|---|---|---|---|---|---|---|
| Equivalent Initialization | 2.475 | 18.06 | 32.86 | 89.40 | 32.52 | 49.91 | 67.80 | 36.18 | 32.15 | 48.69 |
| Normal Initialization | 2.478 | 18.33 | 32.87 | 89.10 | 32.47 | 49.46 | 66.79 | 35.15 | 31.40 | 48.18 |
| Iterative | 2.474 | 22.27 | 18.09 | 86.60 | 31.03 | 49.68 | 65.61 | 34.39 | 31.54 | 47.45 |

Table 6: Model architecture for dense models.

|  | 332M | 650M | 1.5B | 4B |
|---|---|---|---|---|
| Model Dimension | 1024 | 1536 | 2048 | 2560 |
| FFN Dimension | 4096 | 4096 | 5632 | 9728 |
| Attention heads | 16 | 16 | 16 | 32 |
| Layers | 12 | 16 | 24 | 36 |
| Vocabulary Size | 128000 | 128000 | 128000 | 128000 |
| Weight Tying | False | False | False | False |
| Context Length | 8192 | 8192 | 8192 | 8192 |

**Iterative Fusion with Auxiliary Caches.** This approach introduces an iterative fusion process for the reconstruction layers ($\mathcal{L}_R$). We auxiliary maintain one-layer caches, which are iteratively updated. For the first reconstruction layer ($i = n + 1$), the process is identical to the standard FusedKV, where its key and value caches are computed by fusing the caches from source layers 1 and $n$:

$$\boldsymbol{K}^{n+1} = \boldsymbol{a}_{n,1} \odot \boldsymbol{K}^1 + \boldsymbol{a}_{n+1,n} \odot \boldsymbol{K}^n$$
$$\boldsymbol{V}^{n+1} = \boldsymbol{b}_{n+1,1} \odot \boldsymbol{V}^1 + \boldsymbol{b}_{n+1,n} \odot \boldsymbol{V}^n$$

For all subsequent reconstruction layers ($i > n + 1$), the key cache $\boldsymbol{K}^i$ is formed by fusing the reconstructed key from the *previous* layer ($\boldsymbol{K}^{i-1}$) with the cache from the last storage layer ($\boldsymbol{K}^n$). Similarly, the value cache $\boldsymbol{V}^i$ is a fusion of the previous layer's value ($\boldsymbol{V}^{i-1}$) and the cache from the first storage layer ($\boldsymbol{V}^1$). This is formulated as:

$$\boldsymbol{K}^i = \boldsymbol{a}_{i,i-1} \odot \boldsymbol{K}^{i-1} + \boldsymbol{a}_{i,n} \odot \boldsymbol{K}^n, \quad i > n + 1$$
$$\boldsymbol{V}^i = \boldsymbol{b}_{i,i-1} \odot \boldsymbol{V}^{i-1} + \boldsymbol{b}_{i,1} \odot \boldsymbol{V}^1, \quad i > n + 1$$

For initialization, all learnable $d$-dimensional weight vectors ($\boldsymbol{a}, \boldsymbol{b}$) are drawn from a standard normal distribution, $\mathcal{N}(0, 1)$. This iterative design creates a dependency chain across reconstruction layers, allowing for progressively refined representations.

**Standard FusedKV with Initialization Variants.** This is the standard fusion method, where each reconstruction layer independently computes its KV cache by directly fusing the caches from source layers 1 and $n$. In this approach, there is no dependency between the reconstruction layers, allowing for simpler, parallel computation during training and inference. For this architecture, we explore two distinct initialization schemes:

- **Normal Initialization.** Each of the $d$-dimensional weight vectors $\boldsymbol{a}_{i1}, \boldsymbol{a}_{in}, \boldsymbol{b}_{i1}, \boldsymbol{b}_{in}$ for every layer $i$ is independently sampled from a standard normal distribution, $\mathcal{N}(0, 1)$.

- **Equivalent Initialization.** This scheme initializes the standard FusedKV by recursively computing its weights to match the output of the iterative method at initialization. First, sample a set of auxiliary weights from a standard normal distribution for all reconstruction layers $i > n$:

$$\boldsymbol{a}'_{i,i-1}, \boldsymbol{a}'_{i,n}, \boldsymbol{b}'_{i,i-1}, \boldsymbol{b}'_{i,1} \sim \mathcal{N}(0, 1)$$

The final model weights for layer $i$ are then deterministically computed as follows:

For the key weights:

$$\boldsymbol{a}_{i,1} = \begin{cases} \boldsymbol{a}'_{n+1,1} & i = n+1 \\ \boldsymbol{a}'_{i,i-1} \odot \boldsymbol{a}_{i-1,1} & i > n+1 \end{cases}$$

$$\boldsymbol{a}_{i,n} = \begin{cases} \boldsymbol{a}'_{n+1,n} & i = n+1 \\ \boldsymbol{a}'_{i,i-1} \odot \boldsymbol{a}_{i-1,n} + \boldsymbol{a}'_{i,n} & i > n+1 \end{cases}$$

For the value weights:

$$\boldsymbol{b}_{i,1} = \begin{cases} \boldsymbol{b}'_{n+1,1} & i = n+1 \\ \boldsymbol{b}'_{i,i-1} \odot \boldsymbol{b}_{i-1,1} + \boldsymbol{b}'_{i,1} & i > n+1 \end{cases}$$

$$\boldsymbol{b}_{i,n} = \begin{cases} \boldsymbol{b}'_{n+1,n} & i = n+1 \\ \boldsymbol{b}'_{i,i-1} \odot \boldsymbol{b}_{i-1,n} & i > n+1 \end{cases}$$

This one-time computation sets the initial values for the model's parameters $(\boldsymbol{a}, \boldsymbol{b})$, which are subsequently treated as independent and learnable.

We explore three distinct training and initialization strategies for FusedKV, a performance comparison of these strategies is presented in Table 5.

## A.4 DETAILED DERIVATION OF ROPE COMPATIBILITY

In this appendix, we provide the full derivation for the attention score when RoPE-transformed vectors are combined with a learnable weight vector. This derivation supports the claim in the main text that symmetric weights are required to preserve relative positional encoding.

Without loss of generality, we consider the computation within the $j$-th 2D subspace of one attention head. The query $\boldsymbol{q}_m$ at position $m$ and key $\boldsymbol{k}_n$ at position $n$ are transformed by RoPE rotation matrices $\mathbf{R}_{m\theta_j}$ and $\mathbf{R}_{n\theta_j}$, respectively. A learnable weight vector $\boldsymbol{w}_j = [w_{2j}, w_{2j+1}]^T$ is applied element-wise to the transformed key. The attention score $A_{n,j}$ is computed as:

$$A_{n,j} = \tilde{\boldsymbol{q}}_{m,j}^T (\boldsymbol{w}_j \odot \tilde{\boldsymbol{k}}_{n,j})$$

where

$$\begin{aligned} \tilde{\boldsymbol{q}}_{m,j} &= \mathbf{R}_{m\theta_j} \boldsymbol{q}_{m,j} \\ &= \begin{pmatrix} \cos(m\theta_j) & -\sin(m\theta_j) \\ \sin(m\theta_j) & \cos(m\theta_j) \end{pmatrix} \begin{pmatrix} q_{m,2j} \\ q_{m,2j+1} \end{pmatrix} \\ &= \begin{pmatrix} q_{m,2j}\cos(m\theta_j) - q_{m,2j+1}\sin(m\theta_j) \\ q_{m,2j}\sin(m\theta_j) + q_{m,2j+1}\cos(m\theta_j) \end{pmatrix} \end{aligned}$$

and

$$\begin{aligned} \tilde{\boldsymbol{k}}_{n,j} &= \mathbf{R}_{n\theta_j} \boldsymbol{k}_{n,j} \\ &= \begin{pmatrix} \cos(n\theta_j) & -\sin(n\theta_j) \\ \sin(n\theta_j) & \cos(n\theta_j) \end{pmatrix} \begin{pmatrix} k_{n,2j} \\ k_{n,2j+1} \end{pmatrix} \\ &= \begin{pmatrix} k_{n,2j}\cos(n\theta_j) - k_{n,2j+1}\sin(n\theta_j) \\ k_{n,2j}\sin(n\theta_j) + k_{n,2j+1}\cos(n\theta_j) \end{pmatrix} \end{aligned}$$

We now expand the dot product:

$$\begin{aligned} A_j &= \begin{pmatrix} q_{m,2j}\cos(m\theta_j) - q_{m,2j+1}\sin(m\theta_j) \\ q_{m,2j}\sin(m\theta_j) + q_{m,2j+1}\cos(m\theta_j) \end{pmatrix}^T \begin{pmatrix} w_{2j}\left(k_{n,2j}\cos(n\theta_j) - k_{n,2j+1}\sin(n\theta_j)\right) \\ w_{2j+1}\left(k_{n,2j}\sin(n\theta_j) + k_{n,2j+1}\cos(n\theta_j)\right) \end{pmatrix} \\ &= (q_{m,2j}\cos(m\theta_j) - q_{m,2j+1}\sin(m\theta_j)) \cdot w_{2j}(k_{n,2j}\cos(n\theta_j) - k_{n,2j+1}\sin(n\theta_j)) + \\ &\quad (q_{m,2j}\sin(m\theta_j) + q_{m,2j+1}\cos(m\theta_j)) \cdot w_{2j+1}(k_{n,2j}\sin(n\theta_j) + k_{n,2j+1}\cos(n\theta_j)) \\ &= q_{m,2j}k_{n,2j}\left(w_{2j}\cos(m\theta_j)\cos(n\theta_j) + w_{2j+1}\sin(m\theta_j)\sin(n\theta_j)\right) + \\ &\quad q_{m,2j+1}k_{n,2j+1}\left(w_{2j}\sin(m\theta_j)\sin(n\theta_j) + w_{2j+1}\cos(m\theta_j)\cos(n\theta_j)\right) + \\ &\quad q_{m,2j}k_{n,2j+1}\left(w_{2j+1}\sin(m\theta_j)\cos(n\theta_j) - w_{2j}\cos(m\theta_j)\sin(n\theta_j)\right) + \\ &\quad q_{m,2j+1}k_{n,2j}\left(w_{2j+1}\cos(m\theta_j)\sin(n\theta_j) - w_{2j}\sin(m\theta_j)\cos(n\theta_j)\right) \end{aligned}$$

Using the identities $\cos(A)\cos(B) = \frac{\cos(A-B)+\cos(A+B)}{2}$ and $\sin(A)\sin(B) = \frac{\cos(A-B)-\cos(A+B)}{2}$, we regroup the terms.

$$w_{2j}\cos(m\theta_j)\cos(n\theta_j) + w_{2j+1}\sin(m\theta_j)\sin(n\theta_j) =$$
$$\frac{w_{2j}+w_{2j+1}}{2}\cos((m-n)\theta_j) + \frac{w_{2j}-w_{2j+1}}{2}\cos((m+n)\theta_j)$$
$$w_{2j}\sin(m\theta_j)\sin(n\theta_j) + w_{2j+1}\cos(m\theta_j)\cos(n\theta_j) =$$
$$\frac{w_{2j}+w_{2j+1}}{2}\cos((m-n)\theta_j) - \frac{w_{2j}-w_{2j+1}}{2}\cos((m+n)\theta_j)$$

Similarly, using $\sin(A)\cos(B) = \frac{\sin(A-B)+\sin(A+B)}{2}$ and $\cos(A)\sin(B) = \frac{-\sin(A-B)+\sin(A+B)}{2}$:

$$w_{2j+1}\sin(m\theta_j)\cos(n\theta_j) - w_{2j}\cos(m\theta_j)\sin(n\theta_j) =$$
$$\frac{w_{2j+1}+w_{2j}}{2}\sin((m-n)\theta_j) + \frac{w_{2j+1}-w_{2j}}{2}\sin((m+n)\theta_j)$$
$$w_{2j+1}\cos(m\theta_j)\sin(n\theta_j) - w_{2j}\sin(m\theta_j)\cos(n\theta_j) =$$
$$-\frac{w_{2j+1}+w_{2j}}{2}\sin((m-n)\theta_j) + \frac{w_{2j+1}-w_{2j}}{2}\sin((m+n)\theta_j)$$

Substituting these back and collecting terms for $(m-n)$ and $(m+n)$ yields the final form:

$$A_j = \quad \frac{w_{2j}+w_{2j+1}}{2}\left[(q_{m,2j}k_{n,2j} + q_{m,2j+1}k_{n,2j+1})\cos((m-n)\theta_j)\right.$$
$$+(q_{m,2j}k_{n,2j+1} - q_{m,2j+1}k_{n,2j})\sin((m-n)\theta_j)\bigg]$$
$$+\frac{w_{2j}-w_{2j+1}}{2}\left[(q_{m,2j}k_{n,2j} - q_{m,2j+1}k_{n,2j+1})\cos((m+n)\theta_j)\right.$$
$$-(q_{m,2j}k_{n,2j+1} + q_{m,2j+1}k_{n,2j})\sin((m+n)\theta_j)\bigg]$$

This final equation clearly shows that the term multiplied by $\frac{w_{2j}-w_{2j+1}}{2}$ depends on the absolute positions through $(m+n)\theta_j$. This term vanishes if and only if $w_{2j} = w_{2j+1}$, leaving only the terms dependent on the relative position $m - n$.

## A.5 ANALYSIS OF TRAINING DYNAMICS

**Models Exhibit Practical Convergence.** To provide a comprehensive understanding of the performance evolution during training, we evaluate checkpoints of our **1.5B-parameter** models at every 40B tokens. The results, presented in Figure 11, allow for a detailed analysis of convergence trends and the sustained effectiveness of our proposed methods. The learning curves across a majority of the benchmarks demonstrate clear diminishing returns. For instance, the Wikitext perplexity curve, as well as the average score across all tasks, begin to flatten in the later stages of training. The performance improvement between 320B and 400B tokens is considerably smaller than in earlier intervals. This pattern indicates that the models are approaching a state of practical convergence, where relative model capabilities can be compared reliably, even if absolute convergence has not been reached. The performance gap between our methods and the baseline does not diminish over time.

**Sustained Advantage at Convergence.** To further validate the sustained advantage of our methods under near-converged conditions, we continued training from the 400B token checkpoint for an additional 200B tokens. During this extended phase, we used a fixed learning rate of 3e-5, matching the final rate of the initial training. The results, presented in Figure 12, show that the performance curves for all models have almost completely flattened, indicating that they have reached a state of convergence. Crucially, both FusedKV and FusedKV-Lite maintain a consistent and significant performance lead over the Vanilla baseline across all benchmarks. This confirms that the performance gap does not diminish even after extensive training, providing strong evidence that our methods yield a fundamentally more capable model, rather than merely accelerating initial convergence.

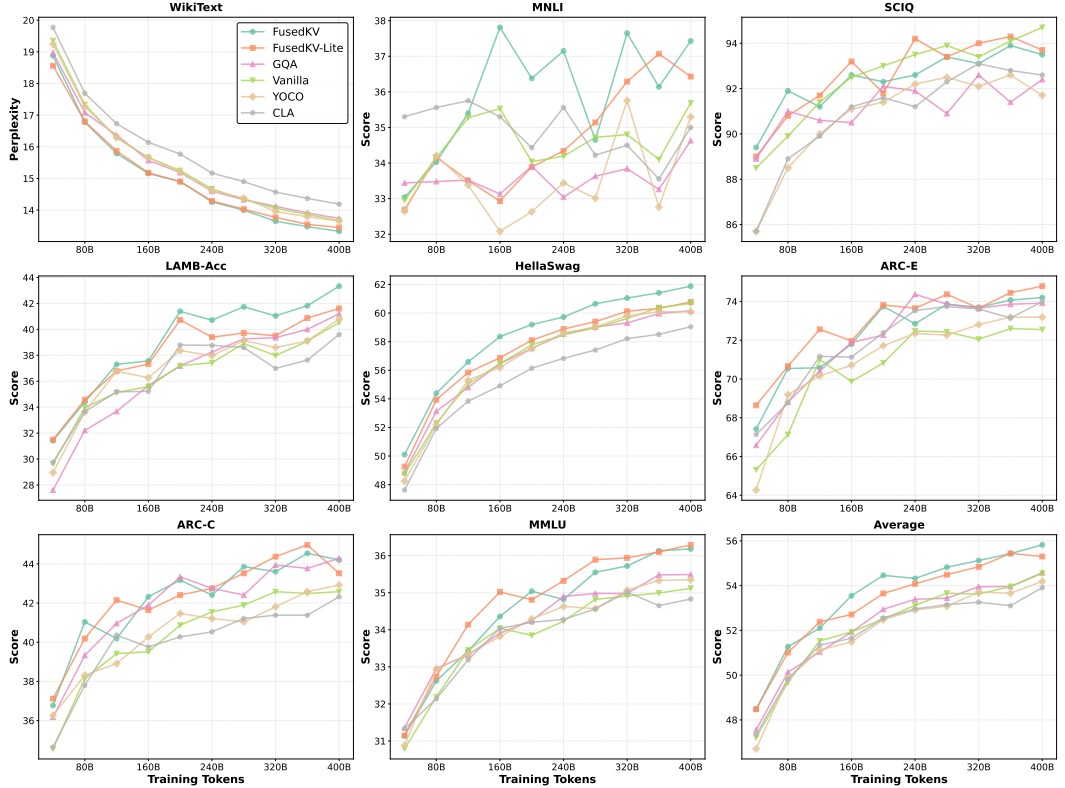

Figure 11: Performance evaluation during final training from 400B to 600B tokens. The models exhibit converged performance, as shown by the stable metric scores. Crucially, the performance gap between our proposed methods (FusedKV and FusedKV-Lite) and the other baselines does not diminish, confirming the long-term effectiveness of our approach.

Table 7: Configuration parameters for the MLA baseline model.

| Parameter | Value |
|---|---|
| attention_heads | 16 |
| q_lora_rank | 864 |
| kv_lora_rank | 736 |
| qk_nope_head_dim | 64 |
| qk_rope_head_dim | 32 |
| v_head_dim | 96 |

## A.6   COMPATIBILITY WITH OTHER ARCHITECTURES

### A.6.1   COMPATIBILITY WITH MLA

While our proposed cross-layer fusion method, FusedKV, offers a novel way to reduce KV cache size by reconstructing information, alternative approaches achieve this through fundamental architectural modifications. A prominent example is Multi-Head Latent Attention (Liu et al., 2024a) , which reduces cache dimensionality by design rather than relying on reconstruction across layers. To investigate whether these two architextures are synergistic, we evaluate the compatibility and combined benefits of applying our method on top of an MLA-based model.

**Experimental Setup.**   We conduct experiments by applying FusedKV to a model already equipped with MLA. Our baseline is a 650M-parameter model, configured with the MLA settings detailed in Table 7. This configuration is designed to halve the cache size compared to a standard Multi-Head Attention (MHA) model while maintaining the same attention computation cost.

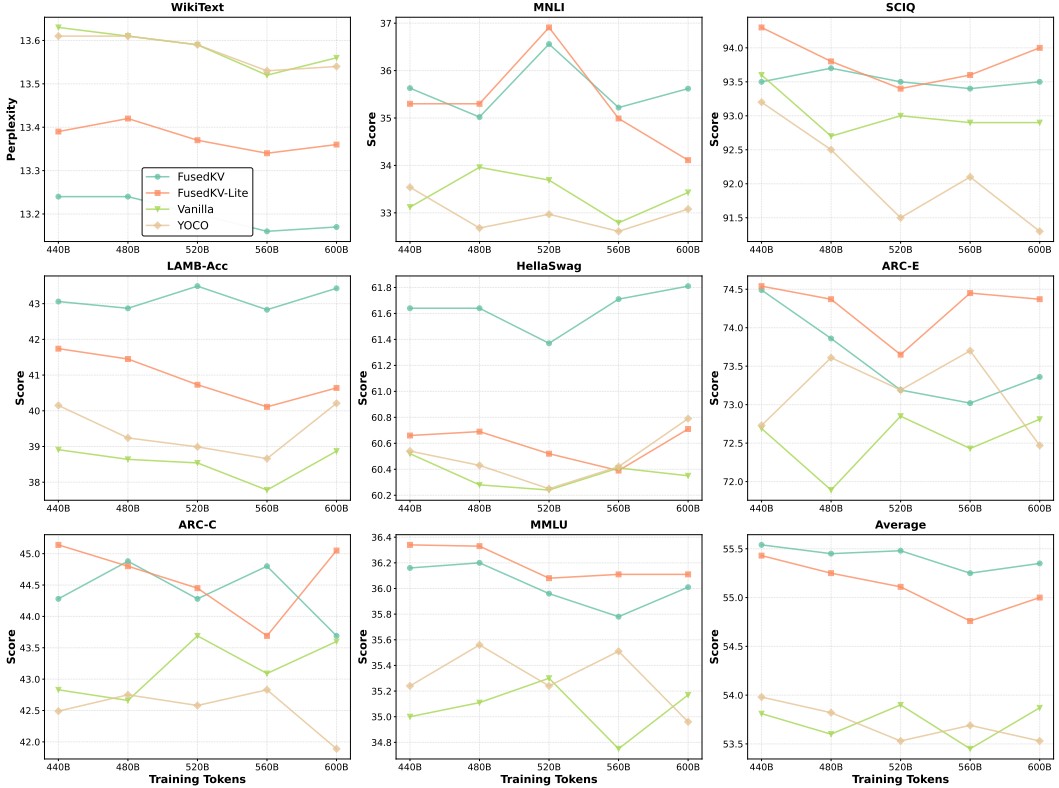

Figure 12: Model performance at convergence (400B to 600B tokens). This extended training run with a fixed learning rate shows that the performance advantage of FusedKV and FusedKV-Lite is maintained even after the models have fully converged.

In our combined setup, denoted as MLA+FusedKV, we apply our fusion mechanism to the MLA architecture. Specifically, for the top half of the layers, we reconstruct the compressed key-value representation (kv_compress) and the key's positional embeddings (k_pos_emb) through a learnable fusion of the correspondings from the bottom and middle layers.

It is worth noting that the standard MLA design jointly compresses K and V into a single kv_compress matrix. This design choice conflicts with the asymmetric sharing mechanism central to our more aggressive optimization, FusedKV-Lite. Adapting MLA to support such asymmetric structures would require non-trivial architectural modifications, which we leave as a promising direction for future work.

**Model Performance and Memory Footprint** We first compare the performance on perplexity and downstream benchmarks. As shown in Table 8, the combination of FusedKV with MLA yields comparable results. MLA+FusedKV achieves a lower perplexity on WikiText compared to the baseline MLA and outperforms it on 5 out of 8 downstream benchmarks. This result demonstrates that our cross-layer fusion method provides benefits even when stacked on top of MLA.

As expected, the baseline MLA model itself incurs a performance drop compared to standard MHA due to the information loss inherent in its KV compression scheme. The results indicate that FusedKV can be a valuable tool to further compress models like MLA, offering practitioners a compelling trade-off between performance and an even smaller memory footprint.

**Inference Speed Analysis.** A direct empirical measurement of inference speed is heavily dependent on highly optimized, hardware-specific kernels. To provide a principled analysis without the substantial engineering effort required for integration into production-grade frameworks like vLLM or SGLang, we estimated the inference speed using the open-source InferSim toolkit Team & Lab.

Table 8: Comparison of performance between MLA and MLA+FusedKV. Memory is shown as a fraction of the MHA model's cache size. FusedKV demonstrates synergistic gains when applied to MLA, enabling further cache reduction with a minimal performance trade-off.

| method | Cache Mem. | Wiki Text↓ | MNLI | SCIQ | LAMB-Acc | Hella Swag | ARC-E | ARC-C | MMLU | Avg Acc↑ |
|---|---|---|---|---|---|---|---|---|---|---|
| MHA | 1 | 18.47 | 34.39 | 88.1 | 31.67 | 49.13 | 65.36 | 36.26 | 31.62 | 48.08 |
| MHA+FusedKV | 1/2 | 18.09 | 33.33 | 86.6 | 31.03 | 49.68 | 65.61 | 34.39 | 31.54 | 47.45 |
| MLA | 1/2 | 21.03 | 33.81 | 85.8 | 27.15 | 43.62 | 62.84 | 32.25 | 29.77 | 45.03 |
| MLA+FusedKV | 1/4 | 20.08 | 32.52 | 80.9 | 28.66 | 45.13 | 62.92 | 33.45 | 29.93 | 44.79 |

Table 9: Comparison of TTFT and TPOT between MLA, MLA+FusedKV, and MHA+FusedKV across varying input lengths for a 16-head model (I/O-bound). Our method significantly reduces prefill time (TTFT) with a small overhead in generation time (TPOT).

| Input/Output Length | TTFT (ms) (MLA / MLA+FusedKV / MHA+FusedKV) | TPOT (ms) (MLA / MLA+FusedKV / MHA+FusedKV) |
|---|---|---|
| 2k/2k | 25.14 / 20.27 / 18.21 | 23.12 / 27.45 / 90.36 |
| 4k/2k | 47.6 / 31.52 / 27.18 | 32.00 / 39.21 / 134.32 |
| 8k/2k | 103.07 / 59.29 / 50.64 | 27.52 / 34.01 / 121.68 |
| 16k/2k | 254.28 / 134.96 / 117.46 | 25.33 / 31.46 / 116.16 |
| 32k/2k | 716.7 / 366.31 / 331.42 | 24.25 / 30.20 / 113.36 |
| 64k/2k | 2288.26 / 1152.37 / 1082.56 | 23.76 / 29.62 / 110.08 |
| 128k/2k | 8010.93 / 4014.26 / 3874.62 | 23.58 / 29.39 / 109.32 |
| 256k/2k | 29774.51 / 14897.16 / 14617.86 | 23.02 / 28.81 / 108.92 |

Table 10: Comparison of TTFT and TPOT for a 48-head model (compute-bound). In this scenario, the I/O overhead of FusedKV is hidden by computation, resulting in comparable TPOT to the baseline MLA while retaining the TTFT advantage.

| Input/Output Length | TTFT (ms) (MLA / MLA+FusedKV) | TPOT (ms) (MLA / MLA+FusedKV) |
|---|---|---|
| 2k/2k | 35.04 / 25.28 | 51.35 / 51.35 |
| 4k/2k | 73.29 / 44.46 | 77.99 / 77.99 |
| 8k/2k | 180.61 / 98.22 | 68.29 / 68.29 |
| 16k/2k | 516.42 / 266.33 | 63.53 / 63.53 |
| 32k/2k | 1671.68 / 844.38 | 61.17 / 61.17 |
| 64k/2k | 5917.23 / 2967.99 | 60.08 / 60.08 |
| 128k/2k | 22147.56 / 11084.81 | 59.64 / 59.64 |
| 256k/2k | 85563.08 / 42795.90 | 58.79 / 58.79 |

(2025). We analyze two key metrics: Time to First Token (TTFT) for the prefill phase and Time Per Output Token (TPOT) for the decoding phase.

I/O-BOUND SCENARIO.    We first analyze a 16-head model configuration, which typically represents a I/O-bound scenario. As detailed in Table 9, applying FusedKV significantly improves prefill speed. MLA+FusedKV reduces TTFT to nearly half that of MLA alone, a benefit stemming from the smaller cache size that needs to be written to memory. For decoding, MLA+FusedKV incurs a minor TPOT increase (approx. 4–6 ms) over MLA. This slight overhead is due to the additional I/O operations required for fetching tensors from multiple layers to perform the fusion step during each token's generation.

COMPUTE-BOUND SCENARIO.    As established in recent work Wang et al. (2025); Chang et al. (2024), when a model's Arithmetic Intensity(AI) exceeds the GPU's compute-to-memory ratio, the latency from I/O communication can be effectively hidden by computation. For MLA, the AI is approximately $4\times$ heads Wang et al. (2025). To demonstrate this principle, we configured a model with 48 attention heads, creating a compute-bound scenario where the AI of MLA+FusedKV surpasses NVIDIA H20 GPU's compute-to-memory ratio. As shown in Table 10, in this setting, the additional I/O latency from our fusion step is fully overlapped by the increased computation. Consequently, while the prefill (TTFT) benefits remain, the decoding speed (TPOT) becomes identical to that of the standard MLA model.

### A.6.2 COMPATIBILITY WITH GQA

Beyond MLA, another widely adopted architectural optimization for improving inference efficiency is Grouped-Query Attention (GQA) Ainslie et al. (2023). To investigate synergistic effects with GQA, we conduct experiments on a 650M-parameter model, combining FusedKV with a GQA architecture. All training parameters were aligned with those used in Table 2. Our results, presented in Table 11, demonstrate that FusedKV is highly compatible with GQA and their combination yields a compelling performance-cost trade-off.

Table 11: Performance comparison demonstrating the synergy between FusedKV and GQA. The combined method FusedKV+GQA achieves a 4x cache reduction and substantial speedup, while mitigating the accuracy loss of GQA alone.

| method | Wiki Text $\downarrow$ | MNLI | SCIQ | LAMB-Acc | Hella Swag | ARC-E | ARC-C | MMLU | Avg Acc$\uparrow$ | TPOT$\downarrow$ | Cache Mem. |
|---|---|---|---|---|---|---|---|---|---|---|---|
| Vanilla (MHA) | 18.47 | 34.39 | 88.1 | 31.67 | 49.13 | 65.36 | 36.26 | 31.62 | 48.08 | 1.000 | 1 |
| GQA | 19.05 | 33.05 | 86.1 | 30.64 | 48.58 | 65.28 | 34.13 | 31.51 | 47.04 | 0.512 | 1/2 |
| FusedKV | 18.09 | 33.33 | 86.6 | 31.03 | 49.68 | 65.61 | 34.39 | 31.54 | 47.45 | 1.408 | 1/2 |
| FusedKV+GQA | 18.48 | 34.11 | 85.3 | 30.16 | 49.04 | 64.87 | 35.67 | 30.92 | 47.15 | 0.741 | 1/4 |

### A.6.3 COMPATIBILITY WITH MOE

**Experimental Setup.** We conducted an experiment on an MoE (Shazeer et al., 2017) model variant. Starting with the 650M-parameter dense model, we replaced its FFN layers with an 8-expert, top-2 gating MoE structure. This model was then trained on a corpus of 100B tokens. Due to the limited model scale and training data available for this specific experiment, our analysis focuses on foundational metrics: training loss and Wikitext perplexity.

**Performance comparision.** The results, presented in Table 12, demonstrate that our methods maintain a strong competitive advantage within the MoE framework. Both FusedKV+MoE and FusedKV-Lite+MoE outperform the cross-layer sharing baseline YOCO+MoE. Notably, FusedKV+MoE achieves the best Wikitext perplexity, while FusedKV-Lite+MoE secures the lowest training loss.

Table 12: Performance comparison on a Mixture-of-Experts (MoE) model. Our methods outperform baseline sharing techniques, demonstrating their applicability to sparse architectures.

| Method | WikiText$\downarrow$ | Train Loss$\downarrow$ |
|---|---|---|
| Vanilla+MoE | 19.62 | 2.513 |
| GQA+MoE | 19.71 | 2.535 |
| YOCO+MoE | 20.17 | 2.544 |
| FusedKV-Lite+MoE | 19.58 | **2.524** |
| FusedKV+MoE | **19.50** | 2.531 |

### A.6.4 COMPATIBILITY WITH SLIDING WINDOW

A prominent hybrid architecture is the integration of Sliding Window Attention (SWA) Jiang et al. (2023) with full attention. In this section, we investigate the adaptability of FusedKV and FusedKV-Lite to a hybrid model that combines both SWA and full attention.

**Experimental Setup.** To explore this, we construct a hybrid architecture based on a 650M-parameter, 16-layer model. The architecture interleaves SWA layers with full attention (FA) layers in a repeating pattern:

- **Window Attention Layers** ($L_w$): $\{1, 2, 3, 5, 6, 7, 9, 10, 11, 13, 14, 15\}$, using a window size of 512.
- **Full Attention Layers** ($L_f$): $\{4, 8, 12, 16\}$.

We design two integration strategies for applying our methods to the top-half layers of the model: **(+W)**: KV cache reconstruction/sharing is applied **only** to the Sliding Window Attention layers.

Table 13: Performance comparison on a hybrid SWA architecture. Best result per column is in bold. Our methods show strong compatibility, significantly reducing cache memory while maintaining or improving the average downstream score.

| Method | Cache Mem. | WikiText↓ | MNLI | SCIQ | LAMB-Acc | HellaSwag | ARC-E | ARC-C | MMLU | Average |
|---|---|---|---|---|---|---|---|---|---|---|
| Baseline (Hybrid) | 29.7% | **24.04** | 34.38 | 87.1 | 31.88 | 49.35 | 65.19 | 35.41 | 31.58 | 47.84 |
| FusedKV-Lite +W | 25.4% | 30.23 | 33.61 | 88.4 | 30.53 | **50.59** | **69.44** | **37.71** | **31.82** | 48.87 |
| FusedKV-Lite +WF | 12.9% | 32.40 | **35.26** | **89.3** | 31.26 | 50.20 | 66.79 | 37.03 | 31.78 | 48.80 |
| FusedKV +W | 25.4% | 28.34 | 33.75 | 86.4 | 32.23 | 50.12 | 68.14 | 36.35 | 31.37 | 48.34 |
| FusedKV +WF | 12.9% | 29.63 | 34.00 | 88.7 | **32.49** | 50.49 | 68.01 | 37.20 | 31.30 | **48.88** |

**(+WF)**: KV cache reconstruction/sharing is applied to **both** Window and Full Attention layers in the top-half layers.The specific rules for the more comprehensive **+WF** strategies are defined as follows:

FUSEDKV-LITE +WF   This strategy employs direct sharing from distinct source layers based on the target layer's type:

$$\text{For SWA layers: } K^i = K^7, \quad V^i = V^1, \qquad \text{if } i > 8 \text{ and } i \in L_w$$
$$\text{For FA layers: } K^i = K^8, \quad V^i = V^4, \qquad \text{if } i > 8 \text{ and } i \in L_f$$

FUSEDKV +WF   This strategy uses learnable fusion, again respecting the layer types:

$$\text{For SWA layers: } K^i = F(K^1, K^7), \quad V^i = F(V^1, V^7), \qquad \text{if } i > 8 \text{ and } i \in L_w$$
$$\text{For FA layers: } K^i = F(K^4, K^8), \quad V^i = F(V^4, V^8), \qquad \text{if } i > 8 \text{ and } i \in L_f$$

**Performance on Hybrid Architecture**   The results, presented in Table 13, demonstrate that both FusedKV and FusedKV-Lite can be effectively adapted to hybrid architectures, yielding not only significant memory savings but also competitive or even superior performance on downstream tasks. While a slight degradation in WikiText perplexity is observed when applying our methods, the average performance across the eight downstream benchmarks is consistently improved. These results strongly suggest that our cross-layer fusion and asymmetric sharing are not limited to uniform architectures but is a flexible and powerful technique that can be tailored to advanced hybrid models.

## A.6.5   COMPATIBILITY WITH OTHER EFFICIENCY TECHNIQUES

A key measure of a new efficiency technique's utility is its ability to compose with existing methods. To evaluate the broader applicability of FusedKV and FusedKV-Lite, we investigate its compatibility with two prevalent model compression strategies: quantization (Liu et al., 2024c) and pruning (Sun et al.).

**Interaction with Quantization**   Quantization reduces memory footprint by representing weights and activations with lower-precision data types. We explored the orthogonality of our method by combining FusedKV and FusedKV-Lite with Kivi (Liu et al., 2024c). The experiments were conducted on a 1.5B-parameter model, applying 4-bit and 2-bit quantization to the KV cache.

The results, presented in Table 14, are highly encouraging. When combined with 4-bit quantization, both FusedKV and FusedKV-Lite maintain their performance with almost no degradation (an average accuracy drop of only  0.2 points). This combination achieves a substantial 4x reduction in the final cache memory footprint on top of the savings from our method itself. This demonstrates that FusedKV and FusedKV-Lite is largely orthogonal to quantization.

**Interaction with Pruning**   Model pruning reduces the number of parameters by removing weights deemed unimportant, which directly translates to lower model storage and potentially faster computation. When combining our methods with Wanda (Sun et al.), we specifically pruned 50% of the Wanda model's parameters using an unstructured approach.

As shown in Table 15, a post-hoc application of pruning to our method resulted in a noticeable degradation in performance across all benchmarks. This suggests that while our method can be combined with common pruning methods, a more sophisticated co-design or joint optimization strategy may be required to mitigate the compounded performance loss.

Table 14: Performance of FusedKV and FusedKV-Lite combined with 2-bit and 4-bit quantization. With 4-bit quantization, performance is maintained while reducing cache memory by 4x.

| Method | WikiText↓ | MNLI | SCIQ | LAMB-Acc | HellaSwag | ARC-E | ARC-C | MMLU | Average | Cache Mem. |
|---|---|---|---|---|---|---|---|---|---|---|
| FusedKV (FP16) | 13.33 | 37.43 | 93.5 | 43.33 | 61.88 | 74.20 | 44.20 | 36.18 | 55.82 | 1 |
| + 4-bit Quant. | 13.39 | 37.37 | 93.4 | 42.73 | 61.78 | 73.86 | 43.94 | 36.30 | 55.63 | 1/4 |
| + 2-bit Quant. | 15.49 | 34.89 | 90.7 | 25.42 | 59.10 | 70.24 | 40.61 | 34.11 | 50.73 | 1/8 |
| FusedKV-Lite (FP16) | 13.45 | 36.43 | 93.7 | 41.61 | 60.77 | 74.79 | 43.52 | 36.29 | 55.30 | 1 |
| + 4-bit Quant. | 13.53 | 36.41 | 93.9 | 40.85 | 60.70 | 74.03 | 43.60 | 36.36 | 55.12 | 1/4 |
| + 2-bit Quant. | 16.33 | 36.16 | 91.5 | 20.05 | 58.71 | 68.94 | 42.58 | 34.03 | 50.28 | 1/8 |

Table 15: Performance of FusedKV and FusedKV-Lite combined with Wanda (50% unstructured pruning). A naive combination leads to a significant performance drop.

| Method | WikiText↓ | MNLI | SCIQ | LAMB-Acc | HellaSwag | ARC-E | ARC-C | MMLU | Average | Model Storage |
|---|---|---|---|---|---|---|---|---|---|---|
| FusedKV | 13.33 | 37.43 | 93.5 | 43.33 | 61.88 | 74.20 | 44.20 | 36.18 | 55.82 | 1 |
| FusedKV + Wanda | 20.63 | 35.42 | 89.9 | 32.76 | 51.09 | 63.85 | 35.75 | 31.18 | 48.56 | 1/2 |
| FusedKV-Lite | 13.45 | 36.43 | 93.7 | 41.61 | 60.77 | 74.79 | 43.52 | 36.29 | 55.30 | 1 |
| FusedKV-Lite + Wanda | 21.12 | 34.57 | 91.3 | 27.50 | 49.59 | 67.51 | 36.09 | 31.51 | 48.30 | 1/2 |

Table 16: Performance comparison on the RULER benchmark for 1.5B-parameter models with a 128k context window. Our methods, FusedKV and FusedKV-Lite, retain a significant portion of the vanilla model's long-context performance, far surpassing the competing compression method YOCO.

| Model | Average | cwe | fwe | niah_multikey_1 | niah_multikey_2 | niah_multiquery | niah_multivalue | niah_single_1 | niah_single_2 | niah_single_3 | qa_hotpotqa | qa_squad | vt |
|---|---|---|---|---|---|---|---|---|---|---|---|---|---|
| Vanilla-128k | 49.11 | 89.0 | 78.67 | 94.0 | 63.0 | 16.0 | 29.0 | 27.0 | 90.0 | 82.0 | 28.0 | 21.0 | 12.8 |
| YOCO-128k | 17.32 | 25.0 | 76.0 | 8.0 | 31.0 | 1.5 | 0.0 | 47.0 | 0.0 | 0.0 | 21.0 | 14.0 | 1.6 |
| FusedKV-Lite-128k | 42.31 | 78.3 | 75.67 | 91.0 | 18.0 | 0.75 | 28.5 | 98.0 | 97.0 | 18.0 | 20.0 | 19.0 | 5.8 |
| FusedKV-128k | 42.00 | 71.3 | 64.0 | 85.0 | 4.0 | 5.25 | 40.25 | 42.0 | 90.0 | 87.0 | 27.0 | 16.0 | 10.2 |

Table 17: Ablation study on key and value source indices for FusedKV-Lite. The top 8 layers reuse caches from the specified source layers. The results confirming our design hypothesis of FusedKV-Lite. Best results in each group are in bold.

| Method | Valid Loss↓ | WikiText↓ | MNLI | SCIQ | LAMB-Acc | HellaSwag | ARC-E | ARC-C | MMLU | Average |
|---|---|---|---|---|---|---|---|---|---|---|
| Vanilla (MHA) | 2.483 | 18.47 | 34.39 | 88.1 | 31.67 | 49.13 | 65.36 | 36.26 | 31.62 | 48.08 |
| *Ablation on Value Source Index (Key Source = Layer 8)* | | | | | | | | | | |
| **value1key8 (FusedKV-Lite)** | **2.473** | **18.55** | 34.24 | 86.9 | **32.95** | **49.53** | **66.58** | **37.46** | **31.88** | **48.51** |
| value5key8 | 2.489 | 18.62 | 34.06 | **88.0** | 31.86 | 48.82 | 66.46 | 37.37 | 30.99 | 48.22 |
| value8key8 (Symmetric) | 2.498 | 19.21 | **34.96** | 86.7 | 30.74 | 48.65 | 64.14 | 34.39 | 30.66 | 47.18 |
| *Ablation on Key Source Index (Value Source = Layer 1)* | | | | | | | | | | |
| **value1key8 (FusedKV-Lite)** | **2.473** | 18.55 | 34.24 | 86.9 | **32.95** | 49.53 | 66.58 | **37.46** | **31.88** | **48.51** |
| value1key5 | **2.473** | **18.40** | **34.36** | **88.4** | 30.25 | **50.21** | **67.09** | 36.69 | 31.72 | 48.39 |
| value1key3 | 2.479 | 18.60 | 34.28 | 87.1 | 31.42 | 49.19 | 66.16 | 36.95 | 30.89 | 48.00 |

## A.7 LONG CONTEXT PERFORMANCE

The primary benefit of KV cache optimization is most pronounced in long-sequence inference scenarios. To rigorously validate the efficacy of our proposed methods under such conditions, we evaluated FusedKV and FusedKV-Lite on models with an extended context window.

**Context Window Extension.** We extended our 1.5B-parameter models to support a 128k context window. We employ a two-stage training strategy. In the first stage, we extend the context length to 32k with 50B tokens, and the RoPE base to 50,000. In the second stage, we further extend the context from 32k to 128k tokens with an additional 25B tokens, and the RoPE base is 500,000.

**Performances.** We evaluate models on the RULER benchmark (Hsieh et al., 2024). The results, presented in Table 16, demonstrate the effectiveness of our approach in preserving long-range dependencies. This stark difference highlights the effectiveness of FusedKV and FusedKV-Lite. While some information loss is inevitable with any form of cache compression, our methods demonstrate a significantly more graceful trade-off, retaining critical long-range information that is essential for practical long-context applications.

### A.8 ABLATION STUDY ON ASYMMETRIC KV CACHE SHARING

A core principle of our FusedKV-Lite method is the hypothesis of asymmetric cache sharing: that the optimal source layers for reconstructing keys and values are different. To systematically validate this hypothesis and quantify the impact of source layer selection, we conducted a detailed ablation study on a 650M-parameter, 16-layer model. For this study, the top 8 layers (layers 9-16) do not generate their own KV caches. Instead, they reconstruct them by sourcing from a single, earlier layer. We independently vary the source layer index for the key and value caches to isolate their respective contributions.

**Importance of Low-Level Value Information.** The first part of the ablation (varying the value source) reveals a clear performance trend: `value1key8 > value5key8 > value8key8`. Sourcing the value cache from the earliest layer (Layer 1) consistently yields the best average performance and validation loss. As the value source layer index increases and approaches the key source index, performance systematically degrades. The fully symmetric configuration (`value8key8`), which is YOCO, shows the most significant performance drop. This confirms that for reconstructing the value cache, which contains content information, sourcing from foundational, lower-indexed layers is demonstrably superior.

**Importance of High-Level Key Information.** The second part of the ablation (varying the key source) shows that performance declines as the key source index decreases: `value1key8 > value1key5 > value1key3`. This indicates that higher-indexed keys, which have undergone more layers of processing and presumably contain richer semantic and positional context, are more informative for reconstructing the attention mechanism in the top layers. However, it is noteworthy that the performance degradation from varying the key source index is less pronounced than when the value source is changed. This observation aligns with our analysis in Figure 2, which suggests that the contribution weights for keys are more diffusely distributed across layers compared to values.

In summary, this ablation study validates that the optimal configuration for KV cache reconstruction is indeed asymmetric. The value cache benefits most from low-level information, while the key cache benefits most from high-level information. Our FusedKV-Lite design, which reuses caches from Layer 1 and Layer 8, represents a well-justified and empirically validated trade-off.

