# OpenReview forum: "Reconstructing KV Caches with Cross-Layer Fusion for Enhanced Transformers"
_ICLR.cc/2026/Conference — ICLR 2026 Poster_

### Official Review · Reviewer_SbV3 · 2025-10-19

**Soundness:** 3
**Presentation:** 3
**Contribution:** 2
**Rating:** 4
**Confidence:** 5

**Summary:**

This paper introduces FusedKV and FusedKV-Lite, two methods for cross-layer KV cache reconstruction in Transformers to reduce memory footprint and prefilling cost. The approach leverages an observed asymmetry between keys and values—where higher-layer keys draw from mid-layer information, while values rely more on lower layers. Both methods halve KV cache memory and achieve 2× acceleration in prefilling while maintaining comparable perplexity to standard Transformers. Experiments across model sizes show consistent performance gains and favorable scaling behavior.

**Strengths:**

1. The evaluation is extensive, covering multiple model scales, diverse tasks, and scaling laws. The inclusion of a scaling experiment convincingly demonstrates the effectiveness and generality of the approach.
2. FusedKV introduces only a small architectural modification, where cross-layer fusion through linear weighting, and achieves measurable efficiency gains. The method integrates smoothly with existing architectures and remains conceptually clear.

**Weaknesses:**

1.	The paper focuses on achieving ~2× cache and prefilling acceleration, while prior methods like YOCO already explore two-stage attention designs (prefill and decode) achieving greater long-sequence efficiency. This limits the novelty in acceleration.
2. Table 2 shows that FusedKV-Lite already provides strong accuracy and latency trade-offs. The heavier FusedKV variant adds learnable parameters and I/O overhead but yields only marginal improvement, making its necessity unclear.
3. Since KV cache architecture primarily affects long-sequence efficiency, the evaluation mainly on perplexity and standard benchmarks doesn’t fully validate the benefit under ultra-long input settings.

**Questions:**

1. Given that FusedKV-Lite already performs well, what is the intended use case for the heavier FusedKV model?
2. How compatible is FusedKV with GQA methods? Can they be combined effectively, and what would be the impact on speedup and accuracy?
3. The paper notes that KV sharing typically reduces parameters in MHA settings. Does the proposed design introduce additional parameters (e.g., in FFN layers or fusion weights), and if so, what is their quantitative effect?

---

> ### Author Response · Authors · 2025-11-22
>
> **Weakness1**
> > The paper focuses on achieving ~2× cache and prefilling acceleration, while prior methods like YOCO already explore two-stage attention designs (prefill and decode) achieving greater long-sequence efficiency. This limits the novelty in acceleration.
>
> Thank you for your insightful feedback.
>
> We agree our work shares the goal of ~2× acceleration with methods like YOCO. However, we wish to emphasize that our core novelty is not the acceleration itself, but the new underlying principle we discovered and the better performance it enables.
>
> 1.  Our key contribution is identifying the asymmetric key-value sharing principle: that upper-layer keys and values are best reconstructed from different source layers.
>
> 2. At the same ~2× prefill acceleration and 50% cache reduction, FusedKV and FusedKV-Lite **significantly outperform YOCO** in model quality (Table 2). We even surpass the full-cache vanilla baseline, a result rarely achieved by prior cache-saving methods. This demonstrates the superiority of our principled approach.
>
> 3. Furthermore, our **FusedKV-Lite** matches the decoding throughput and I/O efficiency of YOCO, while delivering better performance, making it a more ideal and efficient alternative.
>
> In summary, our novelty lies in proposing a new architectural principle and leveraging it to build a model that is more performant at the same level of acceleration. We will revise the paper to highlight these distinctions more clearly.

---

> ### Author Response · Authors · 2025-11-22
>
> **Weakness2 and Question1**
> > Table 2 shows that FusedKV-Lite already provides strong accuracy and latency trade-offs. The heavier FusedKV variant adds learnable parameters and I/O overhead but yields only marginal improvement, making its necessity unclear.
>
> > Given that FusedKV-Lite already performs well, what is the intended use case for the heavier FusedKV model?
>
> Thank you for your insightful feedback.
>
> First, regarding model quality, our findings suggest that the advantages of FusedKV become **more significant at a larger scale**. For instance, in our experiments with a 1.5B-parameter model, FusedKV achieved a lower training loss and demonstrated superior performance on downstream benchmarks. This suggests a valuable path to higher model quality, which is especially critical for larger models.
>
> Second, concerning throughput, we agree that FusedKV's higher I/O makes it less suitable for strictly I/O-bound scenarios. However, in compute-bound scenarios, the additional I/O overhead can be effectively **overlapped** by the computation. As illustrated in Figure 5 (right), this allows FusedKV to achieve a throughput that nearly matches FusedKV-Lite and other methods.
>
> Therefore, we believe FusedKV presents a compelling trade-off: it offers enhanced model quality that scales favorably, while **maintaining competitive throughput in compute-bound environments**. We see it as a promising approach for developing high-performance models.

---

> ### Author Response · Authors · 2025-11-22
>
> **Weakness3**
> > Since KV cache architecture primarily affects long-sequence efficiency, the evaluation mainly on perplexity and standard benchmarks doesn’t fully validate the benefit under ultra-long input settings.
>
>
> Thank you for your valuable suggestion. We agree that evaluating on established long-context benchmarks is crucial to demonstrate the efficacy of KV compression.
>
> Following this principle, we have evaluated our method on the RULER benchmark after extending a 1B model's context window to 128k. As shown ining loss and demonstrated superior performance on downstream benchmarks. This may suggest a valuable path to higher model quality, which is especially critical for larger models.
>
> Second, concerning the table, our proposed FusedKV and FusedKV-Lite methods (**42.0** and **42.31** ) retain the vast majority of the long-context capabilities of the vanilla model (**49.11**). In contrast, another compression technique, YOCO, suffers a drastic performance drop (**17.32**). These results strongly validate our approach's effectiveness in preserving long-range dependencies for practical applications.
>
> | model    | Average  | cwe | fwe | niah_multikey_1 | niah_multikey_2 | niah_multiquery | niah_multivalue | niah_single_1 | niah_single_2 | niah_single_3 | qa_hotpotqa | qa_squad | vt  |
> |-|--|-|-|-|-|-|-|-|-|-|-|-|-|
> | Vanilla-128k|49.11|89 |78.67|94|63|16|29|27|90|82|28|21|12.8|
> | YOCO-128k | 17.32 |25| 76|8| 31|1.5|0| 47 |0|0|21|14|1.6|
> | FusedKV-Lite-128k| 42.31| 78.3| 75.67|91 | 18| 0.75| 28.5| 98 | 97| 18| 20| 19                  | 5.8|
> | FusedKV-128k       | 42          | 71.3           | 64             | 85                         | 4                          | 5.25                       | 40.25                      | 42                       | 90                       | 87                       | 27                     | 16                  | 10.2           |

---

> ### Author Response · Authors · 2025-11-22
>
> **Question2**
> >How compatible is FusedKV with GQA methods? Can they be combined effectively, and what would be the impact on speedup and accuracy?
>
> Thank you for this insightful question regarding the compatibility of FusedKV and GQA.
>
> To investigate this, we conducted experiments on a 650M-parameter model, aligning all training parameters with those in Table 2. Our results demonstrate that FusedKV is highly compatible with GQA and that they can be combined effectively.
>
> In terms of performance, the combined **FusedKV+GQA** model surpasses GQA alone on average accuracy, which indicates that our FusedKV structure provides an additive performance benefit to the GQA. Regarding inference speed, the **FusedKV+GQA** model is significantly faster than the vanilla baseline. While there is a slight accuracy trade-off compared to the vanilla model, the combination achieves a compelling balance, offering a substantial speedup while mitigating the performance degradation typically seen with GQA.
>
> |       Method   |  Wiki Text$\downarrow$ | MNLI  | SCIQ | LAMB-Acc | Hella Swag | ARC-E | ARC-C | MMLU  | Average  |  TPOT$\downarrow$|Cache  Mem. |
> |--------------|----|-------|-------|-------|-------|------|-------|-------|-------|--------| --- |
> | Vanilla             | 18.47 | 31.62 | 31.67 | 36.26 | 88.1 | 49.13 | 65.36 | 34.39 | 48.08  | 1 | 1|
> | GQA            |  19.05 | 31.51 | 30.64 | 34.13 | 86.1 | 48.58 | 65.28 | 33.05 | 47.04  | 0.512 |$1/2$ |
> | Fusedkv        | 18.09 | 31.54 | 31.03 | 34.39 | 86.6 | 49.68 | 65.61 | 33.33 | 47.45  |1.408|$1/2$ |
> | Fusedkv +GQA    |18.48 | 30.92 | 30.16 | 35.67 | 85.3 | 49.04 | 64.87 | 34.11 | 47.15  |0.741|$1/4$ |

---

> ### Author Response · Authors · 2025-11-22
>
> **Question3**
> > The paper notes that KV sharing typically reduces parameters in MHA settings. Does the proposed design introduce additional parameters (e.g., in FFN layers or fusion weights), and if so, what is their quantitative effect?
>
> Thank you for your insightful question regarding the parameter efficiency of our method.
>
> 1. FusedKV-Lite introduces no additional parameters.
>
> 2. FusedKV introduces a small number of learnable fusion weights in the top-half layers of the model.  However, the number of these additional parameters is minimal. The total added parameters for FusedKV amount to $2 * hidden size * num layers$.
>
> To provide a clear quantitative analysis, we compared the number of trainable parameters for a 1.5B-scale model. The results are shown below:
> | Method| Trainable Params.|
> |------------- | ------------------- |
> | Vanilla| 1,833,017,344 |
> | FusedKV-Lite | **1,732,352,512** |
> | FusedKV | 1,732,426,240 |

---

> ### Comment · Reviewer_SbV3 · 2025-11-24
>
> Dear authors:
>
> I understand that your motivation is for better performance instead of acceleration. Given the research scope of the paper, I am willing to raise my score to 6.
>
> I'd like to discuss the further impact of the paper. The paper focuses on improving the performance of the standard Transformer. I'm curious whether it is compatible with hybrid models for acceleration on decoding and better KV cache management. When we take linear attention or sliding window attention into consideration, the direct fusion of the previous KV cache will be non-trivial.

---

> > ### Author Response · Authors · 2025-11-29
> >
> > >I'd like to discuss the further impact of the paper. The paper focuses on improving the performance of the standard Transformer. I'm curious whether it is compatible with hybrid models for acceleration on decoding and better KV cache management. When we take linear attention or sliding window attention into consideration, the direct fusion of the previous KV cache will be non-trivial.
> >
> > Thank you for your insightful question regarding compatibility with hybrid models.
> >
> > To investigate this, we conducted experiments on a hybrid architecture combining both sliding window attention (SWA) and full attention. We used a 650M-parameter, 16-layer model. The architecture integrates SWA (window size = 512) and full attention in a repeating pattern as follows:
> > 1. Window Attention Layers ($L_w$): $\\{ 1, 2, 3, 5, 6, 7, 9, 10, 11, 13, 14, 15 \\}$
> > 2. Full Attention Layers ($L_f$): $\\{ 4, 8, 12, 16 \\}$
> >
> >
> > We evaluated two integration strategies for FusedKV and FusedKV-Lite:
> > 1. **(+W):** Reconstructing/sharing KV Caches only for SWA layers.
> > 2. **(+WF):** Reconstructing/sharing KV Caches for both SWA and FA layers.
> >
> > `FusedKV-Lite +W`:  $\color{blue}K^{ i } = K^{7},\ $$\color{blue}V^{i} = V^{1},\ \$$\color{blue}i > 8 \ and \ i \in L_w$.
> >
> > `FusedKV-Lite +WF`: $\color{blue}K^{ i } = K^{7},\$$\color{blue}V^{i} = V^{1}\ \$$\color{blue}i > 8 \ and \ i \in L_w \ $;  $\color{purple}K^{ i } = K^{8},\$$\color{purple}V^{i} = V^{4},\$$\color{purple}i > 8 \ and \ i \in L_f$.
> >
> >
> > `FusedKV +W`:   $\color{blue}K^{ i } = F(K^{1},K^{7}),\$$\color{blue}V^{i} = F(V^{1},V^{7}),\ \$$\color{blue}i > 8 \ and \ i \in L_w.$
> >
> > `FusedKV +WF`: $\color{blue}K^{ i } = F(K^{1},K^{7}),\$$\color{blue}V^{i} = F(V^{1},V^{7}),\ \$$\color{blue}i > 8 \ and \ i \in L_w \ $;  $\color{purple}K^{ i } = F(K^{4},K^{8}),\$$\color{purple}V^{i} = F(V^{4},V^{8}),\$$\color{purple}i > 8 \ and \ i \in L_f$.
> >
> > |             Method        |Cache  Mem.     | Wiki Text $\downarrow$ | MNLI  | SCIQ | LAMB-Acc | Hella Swag | ARC-E | ARC-C | MMLU  | average  |
> > |---------------------|----|------------|----------|-------|------|--------------------|-----------|----------|---------------|-------------------|
> > | baseline     |  29.7\% | **24.04** | 34.38 | 87.1 | 31.88 | 49.35 | 65.19 | 35.41 | 31.58 | 47.84  |
> > | FusedKV-Lite +W  |25.4\%| 30.23 | 33.61 | 88.4 | 30.53 | **50.59** | **69.44** | **37.71** | **31.82** | 48.87  |
> > | FusedKV-Lite +WF |**12.9\%**| 32.4  | **35.26** | **89.3** | 31.26 | 50.2  | 66.79 | 37.03 | 31.78 | 48.8   |
> > | FusedKV +W     | 25.4\%| 28.34 | 33.75 | 86.4 | 32.23 | 50.12 | 68.14 | 36.35 | 31.37 | 48.34  |
> > | FusedKV +WF    |**12.9\%**| 29.63 | 34.00    | 88.7 | **32.49** | 50.49 | 68.01 | 37.2  | 31.3  | **48.88**  |
> >
> > The results demonstrate that applying FusedKV and FusedKV-Lite to hybrid architectures yields performance gains across several benchmarks, resulting in a higher average score compared to the baseline. Significantly, the **+WF** strategy reduces KV cache memory usage to **12.9%**. While this reduction theoretically supports improved decoding speed, detailed acceleration profiling is outside the scope of this revision and is reserved for future work.

---

### Official Review · Reviewer_7x2j · 2025-10-25

**Soundness:** 3
**Presentation:** 2
**Contribution:** 2
**Rating:** 4
**Confidence:** 4

**Summary:**

This work proposes a training-based KV cache compression method that approximates top-layer caches by fusing caches from bottom layers. Two options are proposed: FusedKV, which creates a weighted sum of bottom and middle-layer caches to achieve better quality at the cost of some inference overhead, and FusedKV-Lite, which directly reuses a cache from a single, lower layer. The experiments are conducted on a range of models from 300M to 4B parameters, using training budgets on the order of 100B tokens.

**Strengths:**

* The proposed method reduces KV cache memory usage by 50% without any performance drop relative to the baseline.

* The observation of KV cache asymmetry appears to be novel within the compression literature.

**Weaknesses:**

* The overall approach lacks conceptual novelty. The idea of cache reuse has been explored in several prior works [1, 2] in a post-training setting, where cache similarity was used as the criterion for reuse. Approaches based on linear predictors [3] have achieved 4x compression with no performance drop and up to 8x compression with only minor degradation.

* The idea of a weighted fusion of previous keys and values also appeared in [4] in the context of improving gradient flow. It is possible that the improvement in validation loss observed in this work stems from the same mechanism.

* The proposed approach requires costly training from scratch, whereas numerous inexpensive post-hoc techniques already exist that achieve high KV cache compression rates [7, 8] with little to no performance decline (see 1st weakness).

* The most practical application for KV compression techniques is in long-context tasks, where the KV cache incurs significant memory overhead and increases inference latency. To fully demonstrate the technique's efficacy, the authors should provide results on established long-context benchmarks, such as LongBench [5] or RULER [6].

* *(minor)* For the smaller models, performance on benchmarks like MMLU (25% random guess) and ARC-C (25% random guess) is only slightly above chance. Given the high noise and potential for random results, along with the cost of estimating standard deviation, the authors might consider removing these tasks for the smaller models.

References

---

[1] Yang Y. et al. Kvsharer: Efficient inference via layer-wise dissimilar kv cache sharing //arXiv preprint arXiv:2410.18517. – 2024.

[2] Liu A. et al. Minicache: Kv cache compression in depth dimension for large language models //Advances in Neural Information Processing Systems. – 2024. – Т. 37. – С. 139997-140031.

[3] Shutova, Alina, et al. "Cache Me If You Must: Adaptive Key-Value Quantization for Large Language Models." arXiv preprint arXiv:2501.19392 (2025).

[4] Pagliardini M. et al. Denseformer: Enhancing information flow in transformers via depth weighted averaging //Advances in neural information processing systems. – 2024. – Т. 37. – С. 136479-136508.

[5] Bai Y. et al. Longbench: A bilingual, multitask benchmark for long context understanding //arXiv preprint arXiv:2308.14508. – 2023.

[6] Hsieh, Cheng-Ping, et al. "RULER: What's the Real Context Size of Your Long-Context Language Models?." arXiv preprint arXiv:2404.06654 (2024).

[7] Liu, Zirui, et al. "Kivi: A tuning-free asymmetric 2bit quantization for kv cache." arXiv preprint arXiv:2402.02750 (2024).

[8] Hooper, Coleman, et al. "Kvquant: Towards 10 million context length llm inference with kv cache quantization." Advances in Neural Information Processing Systems 37 (2024): 1270-1303.

**Questions:**

* How well would the proposed method perform in a post-training setting? For example, could one take a pre-trained model and simply estimate the fusion coefficients for FusedKV (or directly reuse bottom-layer caches for FusedKV-Lite) without full retraining?

* Do I understand correctly that the trained models use Grouped-Query Attention (GQA) with 16 query heads and 2 KV heads? Or is the number of query heads and KV heads the same in experiments?

> For evaluation, we adopt GQA with 128 query heads and 2 key–value heads as the baseline

* What is the head dimension? This configuration is unusual, as large models typically have 8 or more KV heads. A setup with only 2 KV heads is more common for smaller models (e.g., those with 16 or 32 query heads).

---

> ### Author Response · Authors · 2025-11-22
>
> **Weakness1**
> >  The overall approach lacks conceptual novelty. The idea of cache reuse has been explored in several prior works [1, 2] in a post-training setting, where cache similarity was used as the criterion for reuse. Approaches based on linear predictors [3] have achieved 4x compression with no performance drop and up to 8x compression with only minor degradation.
>
> We thank the reviewer for their insightful comments and for highlighting these related works. We agree that cache reuse is an established concept for model compression. However, our primary contribution lies not in the general idea of reuse, but in a novel, principled approach to cross-layer cache reconstruction that leads to unique performance benefits.
>
> 1. **Conceptual Novelty from Key-Value Asymmetry:**  While prior works [1, 2] often treat the KV cache as a unified block for reuse based on similarity, our work is the first to identify and leverage a critical  **Key-Value Asymmetry**. As our analysis in Figure 2 demonstrates, top-layer Values are predominantly derived from bottom layers, whereas Keys require information from middle layers to facilitate effective information flow. By treating Keys and Values distinctly according to their informational needs, our method introduces a more principled and effective fusion strategy, which is a key conceptual departure from prior works that overlook these different roles.
>
> 2.  **Performance Improvement Beyond Compression:**  A crucial distinction of our work is that it not only compresses the KV cache but also enhances model performance. Most existing cache reuse methods, including those mentioned, typically report a performance trade-off, aiming to minimize degradation. In contrast, our proposed FusedKV and FusedKV-Lite consistently  **outperform the full-cache vanilla baseline**  in both perplexity and downstream task accuracy (as shown in Tables 2 & 3. This demonstrates that FusedKV and FusedKV-Lite are more than a compression technique, they are also architectural enhancement that improves model quality.

---

> ### Author Response · Authors · 2025-11-22
>
> **Weakness2**
> > The idea of a weighted fusion of previous keys and values also appeared in [4] in the context of improving gradient flow. It is possible that the improvement in validation loss observed in this work stems from the same mechanism.
>
> Thank you for your insightful comment. We would like to clarify a key distinction between our work and prior methods that use weighted fusion.
>
> While DenseFormer also employs a fusion mechanism, it operates on the  **output hidden states**  of previous layers. This approach inherently necessitates storing these intermediate hidden states, leading to  **significant additional memory overhead**.
>
> In contrast, our FusedKV method performs a weighted fusion directly on the  **keys and values**  within the attention mechanism. This fundamental difference is not merely a design choice but the source of a unique and significant advantage: instead of adding to the memory burden, it actively  **reduces**  the memory footprint of the KV cache during inference.
>
> Additionally, we agree that FusedKV and FusedKV-Lite improve gradient flow, which is one of the intended benefits. We have included Figure 9 in the revised manuscript, which demonstrates that both methods allow for stronger gradient flow to propagate to lower layers compared to the baselines.
>
> Thus, the core advantage of FusedKV and FusedKV-Lite lies in their synergistic effect: they simultaneously enhance model performance through improved information flow while significantly reducing the memory footprint of the KV cache, achieving a rare combination of efficiency and effectiveness.

---

> ### Author Response · Authors · 2025-11-22
>
> **Weakness3**
> > The proposed approach requires costly training from scratch, whereas numerous inexpensive post-hoc techniques already exist that achieve high KV cache compression rates [7, 8] with little to no performance decline (see 1st weakness).
>
> Thank you for your insightful feedback. We agree that post-hoc quantization methods are highly effective.
>
> We would like to clarify that the cited works focus on  **post-hoc quantization**  of KV caches. Our approaches, FusedKV and FusedKV-Lite, are orthogonal to these methods as they reduces the number of cache layers through asymmetric sharing and cross-layer fusion.
>
> This orthogonality means our method are complementary to, rather than a competitor of, post-hoc techniques. As shown in Table 1 and Table 2, FusedKV/FusedKV-Lite with 4-bit quantization achieves a 4x memory reduction with **almost no degradation in performance**.
>
> **Table 1.**  Performance of FusedKV with 2bit and 4bit quantization.
> |       Method      | Wiki Text$\downarrow$ | MNLI  | SCIQ | LAMB-Acc | Hella Swag | ARC-E | ARC-C | MMLU  | average  | Cache   Mem.|
> |---------|------------------|-------|------|----------|-----------|-------|-------|-------|----------| - |
> | FusedKV           | 13.33 | 37.43 | 93.5 | 43.33 | 61.88 | 74.2  | 44.2  | 36.18 | 55.82  | 1 |
> | +4bit      | 13.39 | 37.37 | 93.4 | 42.73 | 61.78 | 73.86 | 43.94 | 36.3  | 55.63  | $1/4$|
> | +2bit      | 15.49 | 34.89 | 90.7 | 25.42 | 59.1  | 70.24 | 40.61 | 34.11 | 50.73  | $1/8$|
>
> **Table 2.** Performance of FusedKV-Lite with 2bit and 4bit quantization.
> |       Method      | Wiki Text$\downarrow$ | MNLI  | SCIQ | LAMB-Acc | Hella Swag | ARC-E | ARC-C | MMLU  | average  | Cache   Mem.|
> |---------|------------------|-------|------|----------|-----------|-------|-------|-------|----------| - |
> | FusedKV-Lite      | 13.45 | 36.43 | 93.7 | 41.61 | 60.77 | 74.79 | 43.52 | 36.29 | 55.3   | 1 |
> | +4bit | 13.53 | 36.41 | 93.9 | 40.85 | 60.7  | 74.03 | 43.6  | 36.36 | 55.12  |$1/4$|
> | +2bit | 16.33 | 36.16 | 91.5 | 20.05 | 58.71 | 68.94 | 42.58 | 34.03 | 50.28  |$1/8$|
>
> This result highlights that FusedKV and FusedKV-Lite can be powerfully combined with existing post-hoc quantization compression techniques for compounded gains. We will explicitly state this complementary relationship in our revised manuscript.

---

> ### Author Response · Authors · 2025-11-22
>
> **Weakness4**
> > The most practical application for KV compression techniques is in long-context tasks, where the KV cache incurs significant memory overhead and increases inference latency. To fully demonstrate the technique's efficacy, the authors should provide results on established long-context benchmarks, such as LongBench or RULER .
>
>
> Thank you for your valuable suggestion. We agree that evaluating on established long-context benchmarks is crucial to demonstrate the efficacy of KV compression.
>
> Following this principle, we have evaluated our method on the RULER benchmark after extending a 1B model's context window to 128k. As shown in the table, our proposed FusedKV and FusedKV-Lite methods (**42.0** and **42.31** ) retain the vast majority of the long-context capabilities of the vanilla model (**49.11**). In contrast, another compression technique, YOCO, suffers a drastic performance drop (**17.32**). These results strongly validate our approach's effectiveness in preserving long-range dependencies for practical applications.
>
>
> | model    | Average  | cwe | fwe | niah_multikey_1 | niah_multikey_2 | niah_multiquery | niah_multivalue | niah_single_1 | niah_single_2 | niah_single_3 | qa_hotpotqa | qa_squad | vt  |
> |-|-|-|-|-|---|--|-|--|-----|--|--|--|--|
> | Vanilla-128k  | **49.11** | **89**    | **78.67**          | **94**                         | **63** | **16**    | 29    | 27    | 90      | 82   | **28**   | **21**    | **12.8**      |
> | YOCO-128k | 17.32 | 25     | 76     | 8    | 31| 1.5  | 0   | 47  | 0  | 0    | 21     | 14   | 1.6  |
> | FusedKV-Lite-128k| 42.31| 78.3| 75.67  | 91  | 18    | 0.75     | 28.5    | **98**   | **97**    | 18  | 20    | 19 | 5.8 |
> | FusedKV-128k   | 42    | 71.3  | 64    | 85      | 4                          | 5.25     | **40.25**  | 42        | 90  |**87**      | 27                     | 16       | 10.2    |

---

> ### Author Response · Authors · 2025-11-22
>
> **Weakness5**
> > (minor) For the smaller models, performance on benchmarks like MMLU (25% random guess) and ARC-C (25% random guess) is only slightly above chance. Given the high noise and potential for random results, along with the cost of estimating standard deviation, the authors might consider removing these tasks for the smaller models.
>
> Thank you for this valuable and constructive suggestion.
>
> To ensure the comprehensiveness and transparency of our evaluation, we reported results on the same set of tasks across all model sizes. We will add a note to the paper cautioning readers about the potential for high variance in these results for smaller models.

---

> ### Author Response · Authors · 2025-11-22
>
> **Question1**
> >   How well would the proposed method perform in a post-training setting? For example, could one take a pre-trained model and simply estimate the fusion coefficients for FusedKV (or directly reuse bottom-layer caches for FusedKV-Lite) without full retraining?
>
> Thank you for this insightful question. We explore this scenario by applying FusedKV and FusedKV-Lite to a pre-trained 650M model, followed by a 4-epoch fine-tuning stage using around 1B tokens.
>
> |Method| Train loss |
> |-|-|
> |Vanilla| 0.817|
> |FusedKV-Lite| 1.373 |
> |FusedKV|1.372|
>
>
> Our results show that this brief fine-tuning period is insufficient for the model to fully adapt to the modified KV cache structure, as evidenced by the higher training loss compared to the vanilla baseline. However, we expect that the model's capabilities can be more recovered with more training tokens during mid-training stage.

---

> ### Author Response · Authors · 2025-11-22
>
> **Question2**
> >  Do I understand correctly that the trained models use Grouped-Query Attention (GQA) with 16 query heads and 2 KV heads? Or is the number of query heads and KV heads the same in experiments?
>
> Thank you for the clarifying question.
> In our experiments for FusedKV and FusedKV-Lite, the number of query heads and KV heads are the same. The resulting KV cache size is equivalent to that of a GQA model with 16 query and 2 KV heads, which is also $1/2$ of the vanilla.

---

> ### Author Response · Authors · 2025-11-22
>
> **Question3**
> > What is the head dimension? This configuration is unusual, as large models typically have 8 or more KV heads. A setup with only 2 KV heads is more common for smaller models (e.g., those with 16 or 32 query heads).
>
>
> Thank you for your insightful question regarding the model configuration.
>
> The head dimension used in our experiments is 128.
>
> This setup is intentionally chosen to specifically evaluate the inference speed of our proposed FusedKV under a compute-bound scenario.
> When the ratio of query heads to KV heads is very high (64 in Section 2.5), the workload becomes compute-bound on H20 GPU. In such conditions, the I/O time of the small KV Cache could be effectively overlapped by the computational workload. This specific experimental design allows us to demonstrate that, in compute-bound environments, FusedKV **incurs no additional inference latency** compared to the baseline.
>
> We also admit that the precise ratio required to achieve a compute-bound state can vary depending on the specific computational and communication capabilities of the GPU.

---

> > ### Comment · Reviewer_7x2j · 2025-11-22
> >
> > Thank you for your response. Most of my concerns have been addressed. The new results provide additional evidence of the usefulness of the method. Consequently, I have decided to raise my score.

---

### Official Review · Reviewer_K5c9 · 2025-10-28

**Soundness:** 2
**Presentation:** 2
**Contribution:** 2
**Rating:** 4
**Confidence:** 4

**Summary:**

This paper proposes a new KV cache sharing algorithm, FusedKV. By leveraging their findings on key-value reconstructions, they suggested which layers should be connected for sharing kv pairs. This ends up sharing values from bottom layers and keys from middle layers. They tested two variants, with and without learnable weights.

**Strengths:**

- Drawing insights from preliminary experiment (dense fusion) seems a good research direction to build up their own methods.
- Measure various metrics to compare methods.
- Validation at scale (larger model sizes or larger training token numbers) seems good.

**Weaknesses:**

- I feel like FusedKV is not a good approach due to its lower throughput, which comes from its higher KV IO, and marginal quality differences over FusedKV-Lite.
- Proposed method is too heuristic. Although I personally like heuristic, simple yet effective method, I'm not sure about the generalizability of this proposed method. Using the first and middle layers as the source layer can be not optimal for other architectures (or with other modality). It seems like well-tailored, specific sharing strategy for this scenario.
- In Figure 9, why do you think dense fusion outperforms vanilla models? Don't they (vanilla) have larger degree of freedom (i.e., larger trainable parameters) than reconstructing caches?
- In L.413-414, isn't this typo? Rev means K^i = K^1, V^i = V^8?
- I believe overall presentation could be enhanced, like Figure captions or visualization of results).

**Questions:**

See above Weakness parts.

---

> ### Author Response · Authors · 2025-11-22
>
> **Weakness1**
> >  I feel like FusedKV is not a good approach due to its lower throughput, which comes from its higher KV IO, and marginal quality differences over FusedKV-Lite.
>
> Thank you for your insightful feedback.
>
> First, regarding model quality, our findings suggest that the advantages of FusedKV become **more significant at a larger scale**. For instance, in our experiments with a 1.5B-parameter model, FusedKV achieved a lower training loss and demonstrated superior performance on downstream benchmarks. This suggests a valuable path to higher model quality, which is especially critical for larger models.
>
> Second, concerning throughput, we agree that FusedKV's higher I/O makes it less suitable for strictly I/O-bound scenarios. However, in compute-bound scenarios, which are very common in practice, the additional I/O overhead can be effectively overlapped by the computation. As illustrated in Figure 5 (right), this allows FusedKV to achieve a throughput that **nearly matches** other baseline methods.
>
> Therefore, we believe FusedKV presents a compelling trade-off: it offers enhanced model quality that scales favorably, while maintaining competitive throughput in compute-bound environments. We see it as a promising approach for developing high-performance models.

---

> ### Author Response · Authors · 2025-11-22
>
> **Weakness2**
> > Proposed method is too heuristic. Although I personally like heuristic, simple yet effective method, I'm not sure about the generalizability of this proposed method. Using the first and middle layers as the source layer can be not optimal for other architectures (or with other modality). It seems like well-tailored, specific sharing strategy for this scenario.
>
> We thank the reviewer for the insightful comment on generalizability.
>
> To test whether our layer sharing strategy is an overly "well-tailored" heuristic, we applied it to a fundamentally different architecture: a sparse Mixture-of-Experts (MoE) model. We modified a 650M dense model by replacing its FFN layers with an 8-expert MoE structure, trained on 100B tokens.
>
> Due to the limited model size and training data, we focus our analysis on training loss and Wikitext perplexity.
>
> The results demonstrate that our methods, FusedKV and FusedKV-Lite, outperform the cross-layer baseline YOCO even in this sparse MoE architecture.
>
> | Method |  Wiki Text$\downarrow$ | Train  Loss$\downarrow$|
> |---------|-----------|---------|
> | Vanilla+MoE          | 19.62 | 2.513|
> | GQA+MoE       | 19.71  | 2.535|
> | YOCO+MoE       | 20.17  | 2.544|
> | FusedKV-Lite+MoE | 19.58  | **2.524**|
> | FusedKV+MoE      | **19.50** | 2.531|
>
> This strong performance on an MoE model demonstrates that our proposed sharing strategy is not well-tailored method but a robust principle that generalizes effectively to different model architectures.

---

> ### Author Response · Authors · 2025-11-22
>
> **Weakness3**
> > In Figure 9, why do you think dense fusion outperforms vanilla models? Don't they (vanilla) have larger degree of freedom (i.e., larger trainable parameters) than reconstructing caches?
>
>
> We thank the reviewer for their insightful question regarding the model's parameterization. While our Dense Fusion approach does reduce the number of trainable parameters by omitting the Key and Value projection matrices in the half latter layers, these parameters account for a modest **5.5%** of the baseline model's total. Therefore, the two architectures remain highly comparable in terms of overall size and capacity.
>
> Therefore, we attribute the superior performance of dense fusion not to its parameter count, but rather to its architectural advantages. By enabling later layers to reconstruct their KV caches from earlier ones, the method establishes a more direct and efficient information flow, which is shown in Figure 9.
>
> **Weakness4**
> >   In L.413-414, isn't this typo? Rev means K^i = K^1, V^i = V^8?
>
> Thank you for catching this typo. We will correct this error in the revised manuscript. We apologize for the oversight.
>
>
> **Weakness5**
> >  I believe overall presentation could be enhanced, like Figure captions or visualization of results).
>
> We appreciate the reviewer's valuable suggestion. We have revised the manuscript to enhance the clarity and interpretability of all figures, including the captions and the visualization of results.

---

> > ### Comment · Reviewer_K5c9 · 2025-11-25
> >
> > Thanks for addressing my concerns. I think this paper has well validated their method using huge computational resources.
> >
> > What I'm still worried about some heuristic nature of this method (though I personally like the simple method, yet effective like this) is that I don't think this can be generally applicable the recent and future architectures.
> > What happened if the model is a hybrid architecture of SWA / linear attention and global attention? I'm not sure learning some linear combination of those caches like FusedKV will always work well.
> >
> > Also, is there any ablation study for changing key and value indices for FusedKV-Lite? In my personal experience, I didn't see that value asymmetry patterns, which makes me to feel like it's not generalizable method.
> >
> > So I'll keep my score at this point (but quite leaning towards neutral for the paper decision).

---

> > > ### Author Response · Authors · 2025-11-29
> > >
> > > > is there any ablation study for changing key and value indices for FusedKV-Lite? In my personal experience, I didn't see that value asymmetry patterns, which makes me to feel like it's not generalizable method.
> > >
> > > We thank the reviewer for their insightful question.
> > >
> > > To empirically validate our design choice, we have conducted a new ablation study on the 650M-parameter, 16-layer model. In this ablation, **the top 8 layers do not generate their own key-value caches**, instead, they source it from earlier layers by systematically varying the source layer indices. The results, as presented in the table below, reinforce our initial hypothesis. We will add this ablation to the final manuscript.
> > >
> > >
> > > ### 1. Ablation on the **Value** Source Index
> > >
> > > To isolate the contribution of the value source, we fix the key cache source to the Layer 8 and vary the source layer for the value cache.
> > >
> > > |             Method             | Valid    Loss$\downarrow$ | Wiki   Text$\downarrow$| MNLI  | SCIQ | LAMB-Acc | Hella Swag | ARC-E | ARC-C | MMLU  | average  |
> > > |---------|------------------|-------|------|----------|-----------|-------|-------|-------|----------| - |
> > > | vanilla                  | 2.483      | **18.47**    | 34.39 | **88.1** | 31.67              | 49.13     | 65.36    | 36.26         | 31.62             | 48.08    |
> > > | value1key8(Fusedkv-Lite) | **2.473**      | 18.55    | 34.24 | 86.9 | **32.95**             | **49.53**     | **66.58**    | **37.46**         | **31.88**             | **48.51**    |
> > > | value5key8               | 2.489      | 18.62    | 34.06 | 88   | 31.86              | 48.82     | 66.46    | 37.37         | 30.99             | 48.22    |
> > > | value8key8(YOCO)         | 2.498      | 19.21    | **34.96** | 86.7 | 30.74              | 48.65     | 64.14    | 34.39         | 30.66             | 47.18    |
> > >
> > >
> > > **Peformance: `value1key8`>`value5key8`>`value8key8`**.
> > >
> > > This performance provides strong empirical evidence for our hypothesis: asymmetric sharing is critical, and **for the value cache, lower-indexed layers are demonstrably superior.**
> > >
> > >
> > >
> > > ### 2. Ablation on the **Key** Source Index
> > >
> > > To isolate the contribution of the key source, we fix the value cache source to the Layer 1 and vary the source layer for the key cache.
> > >
> > > | Method      | Valid      Loss$\downarrow$| Wiki  Text$\downarrow$| MNLI  | SCIQ | LAMB-Acc | Hella Swag | ARC-E | ARC-C | MMLU  | average  |
> > > |---------|------------------|-------|------|----------|-----------|-------|-------|-------|----------| - |
> > > | vanilla                  | 2.483      | 18.47    | **34.39** | 88.1 | 31.67              | 49.13     | 65.36    | 36.26         | 31.62             | 48.08    |
> > > | value1key8(Fusedkv-Lite) | **2.473**      | 18.55    | 34.24 | 86.9 | **32.95**              | 49.53     | 66.58    | **37.46**         | **31.88**             | **48.51**    |
> > > | value1key5               | **2.473**      | 18.40     | 34.36 | **88.4** | 30.25              | **50.21**     | **67.09**    | 36.69         | 31.72             | 48.39    |
> > > | value1key3               | 2.479      | **18.60**     | 34.28 | 87.1 | 31.42              | 49.19     | 66.16    | 36.95         | 30.89             | 48.00       |
> > >
> > > **Peformance: `value1key8`>`value1key5`>`value1key3`**.
> > >
> > > Notably, the performance degradation from varying the key source index is less pronounced compared to when the value source is changed. This observation aligns with our findings in Figure 2, which indicate that the contribution weights for keys are more diffusely distributed across layers. Nevertheless, **the consistent decline in performance as the key index decreases** demonstrates that higher-indexed keys are more informative and play a more significant role in reconstructing top-layer attention mechanisms.
> > >
> > > The model architectures used in our paper are quite similar to `Qwen3`, a widely adopted model. We do not introduce any additional structural components, which demonstrates our method's seamless applicability to mainstream model architectures. Furthermore, we have also validated its effectiveness on other diverse structures, such as Mixture-of-Experts (MoE) and Mixture-of-LoRA-Experts (MLA), confirming its robustness across different designs.

---

> > > ### Author Response · Authors · 2025-11-29
> > >
> > > > What I'm still worried about some heuristic nature of this method (though I personally like the simple method, yet effective like this) is that I don't think this can be generally applicable the recent and future architectures. What happened if the model is a hybrid architecture of SWA / linear attention and global attention? I'm not sure learning some linear combination of those caches like FusedKV will always work well.
> > >
> > > We thank the reviewer for this insightful question regarding the generalizability of our method to hybrid architectures. This is a crucial point, and we agree that demonstrating adaptability is key to the method's practical value for future models.
> > >
> > > To investigate this, we conducted experiments on a hybrid architecture combining both sliding window attention (SWA) and full attention. We used a 650M-parameter, 16-layer model. The architecture integrates SWA (window size = 512) and full attention in a repeating pattern as follows:
> > > 1. Window Attention Layers ($L_w$): $\\{ 1, 2, 3, 5, 6, 7, 9, 10, 11, 13, 14, 15 \\}$
> > > 2. Full Attention Layers ($L_f$): $\\{ 4, 8, 12, 16 \\}$
> > >
> > >
> > > We evaluated two integration strategies for FusedKV and FusedKV-Lite:
> > > 1. **(+W):** Reconstructing/sharing KV Caches only for SWA layers.
> > > 2. **(+WF):** Reconstructing/sharing KV Caches for both SWA and FA layers.
> > >
> > > `FusedKV-Lite +W`:  $\color{blue}K^{ i } = K^{7},\ $$\color{blue}V^{i} = V^{1},\ \$$\color{blue}i > 8 \ and \ i \in L_w$.
> > >
> > > `FusedKV-Lite +WF`: $\color{blue}K^{ i } = K^{7},\$$\color{blue}V^{i} = V^{1}\ \$$\color{blue}i > 8 \ and \ i \in L_w \ $;  $\color{purple}K^{ i } = K^{8},\$$\color{purple}V^{i} = V^{4},\$$\color{purple}i > 8 \ and \ i \in L_f$.
> > >
> > >
> > > `FusedKV +W`:   $\color{blue}K^{ i } = F(K^{1},K^{7}),\$$\color{blue}V^{i} = F(V^{1},V^{7}),\ \$$\color{blue}i > 8 \ and \ i \in L_w.$
> > >
> > > `FusedKV +WF`: $\color{blue}K^{ i } = F(K^{1},K^{7}),\$$\color{blue}V^{i} = F(V^{1},V^{7}),\ \$$\color{blue}i > 8 \ and \ i \in L_w \ $;  $\color{purple}K^{ i } = F(K^{4},K^{8}),\$$\color{purple}V^{i} = F(V^{4},V^{8}),\$$\color{purple}i > 8 \ and \ i \in L_f$.
> > >
> > > |             Method        |Cache  Mem.     | Wiki Text $\downarrow$ | MNLI  | SCIQ | LAMB-Acc | Hella Swag | ARC-E | ARC-C | MMLU  | average  |
> > > |---------------------|----|------------|----------|-------|------|--------------------|-----------|----------|---------------|-------------------|
> > > | baseline     |  29.7\% | **24.04** | 34.38 | 87.1 | 31.88 | 49.35 | 65.19 | 35.41 | 31.58 | 47.84  |
> > > | FusedKV-Lite +W  |25.4\%| 30.23 | 33.61 | 88.4 | 30.53 | **50.59** | **69.44** | **37.71** | **31.82** | 48.87  |
> > > | FusedKV-Lite +WF |**12.9\%**| 32.4  | **35.26** | **89.3** | 31.26 | 50.2  | 66.79 | 37.03 | 31.78 | 48.8   |
> > > | FusedKV +W     | 25.4\%| 28.34 | 33.75 | 86.4 | 32.23 | 50.12 | 68.14 | 36.35 | 31.37 | 48.34  |
> > > | FusedKV +WF    |**12.9\%**| 29.63 | 34.00    | 88.7 | **32.49** | 50.49 | 68.01 | 37.2  | 31.3  | **48.88**  |
> > >
> > > Indeed, the landscape of hybrid architectures, including those with linear attention, is still under active exploration by the community. The results on the SWA/FA model demonstrate that our method is not limited to uniform architectures and can be adapted to handle varied layer types. While further research is needed to tailor the approach for linear attention, **we believe our findings provide a vital experience: that asymmetric sharing and cross-layer fusion could be exploited in complex, non-uniform models**. This opens a promising avenue for future research in this area.

---

### Official Review · Reviewer_j1dt · 2025-11-02

**Soundness:** 3
**Presentation:** 3
**Contribution:** 3
**Rating:** 4
**Confidence:** 3

**Summary:**

This paper proposes FusedKV and FusedKV-Lite as techniques to reduce key-value cache memory in transformer decoder-only models. The key contribution is identifying that top-layer key-value caches can be asymmetrically reconstructed: values are predominantly derived from the bottom layer, while keys benefit from information spanning both bottom and middle layers. FusedKV learns a dimension-wise weighted fusion of caches from these two source layers, operating on post-RoPE keys to preserve relative positional information without recomputation. FusedKV-Lite offers a simpler variant that directly reuses the middle-layer keys and bottom-layer values, eliminating fusion overhead. Across model scales from 332M to 4B parameters, both methods reduce cache memory by 50 percent while achieving lower validation perplexity than standard transformers, establishing an effective cross-layer cache sharing paradigm.

**Strengths:**

- The asymmetric key-value sharing principle is a clear, well-motivated insight grounded in empirical analysis (Figure 2), distinguishing this work from prior indiscriminate cross-layer approaches.
- Solid experimental validation across multiple model sizes (332M to 4B) with consistent gains over baselines. Validation loss curves (Figure 6) demonstrate stability across training.
- Practical design: operating on post-RoPE keys elegantly preserves positional information without computational overhead.
- Comprehensive ablation studies (Figure 8) clearly demonstrate the importance of asymmetry direction and learnable weights.
- Efficient implementation details provided, including Triton kernels and complexity analysis (Table 1).

**Weaknesses:**

- Missing fundamental architectural comparison: This paper does not compare against MLA, which reduces cache dimensionality by design rather than reconstructing across layers. For practitioners optimizing cache size, adopting MLA is a more principled solution than reconstructing caches across standard multi-head attention layers. Direct comparison (accuracy, memory, inference speed) would clarify when cross-layer fusion is preferable to architectural redesign -- and particularly the potential of the combination (do the gains stack?). After all, shouldn't we start improving on top of the best known methods?
- Limited scope of baselines: Missing comparisons to recent token-level or head-level reduction methods such as SpindleKV (2025), Value Residual Learning (2025), and Mixture-of-Recursions (2025), which operate on complementary dimensions.​
- Evaluation restricted to standard language modeling and GLUE-like benchmarks. The most KV-cache intensive scenarios are missing from the evaluation: a) truly long-sequence tasks (>100k tokens); b) multi-modal or encoder-decoder architectures.
- No analysis/discussion of training cost or convergence properties relative to vanilla training. Does learnable fusion increase training time or memory significantly?
- Potential interaction with other efficiency techniques unclear. Can this paper's method be orthogonally combined with quantization, pruning, or token merging for further gains?

**Questions:**

- How does this paper's method compare quantitatively against MLA in terms of accuracy, cache memory footprint, and end-to-end inference speed? Since MLA reduces cache size structurally rather than through reconstruction, this comparison would establish whether cross-layer fusion offers advantages over fundamental architectural changes, or whether practitioners should adopt MLA instead. What are the performance-cost tradeoffs?
- How does this paper's approach interact with token-level or head-level cache reduction? If combined with SpindleKV or head-importance pruning, could further memory savings be achieved without degradation? This clarifies orthogonality versus redundancy with complementary approaches.
- Does performance degrade significantly on extremely long sequences (>100k tokens) or knowledge-intensive retrieval tasks? Stress testing on long-context QA would strengthen robustness claims across diverse inference scenarios.
- What is the training overhead compared to vanilla training? How do learnable fusion weights affect gradient flow, training stability, or convergence speed? Reporting training time, memory, and loss curves would clarify practical costs.
- Can the asymmetric sharing principle generalize to encoder-decoder, mixture-of-experts, or other architectural variants? Current evaluation is limited to dense decoder-only models.

---

> ### Author Response · Authors · 2025-11-22
> **Response to Weakness1 and Question1 (1/2)**
>
> **Weakness1 and Question1**
> > Missing fundamental architectural comparison: This paper does not compare against MLA, which reduces cache dimensionality by design rather than reconstructing across layers. For practitioners optimizing cache size, adopting MLA is a more principled solution than reconstructing caches across standard multi-head attention layers. Direct comparison (accuracy, memory, inference speed) would clarify when cross-layer fusion is preferable to architectural redesign -- and particularly the potential of the combination (do the gains stack?). After all, shouldn't we start improving on top of the best known methods?
>
> >How does this paper's method compare quantitatively against MLA in terms of accuracy, cache memory footprint, and end-to-end inference speed? Since MLA reduces cache size structurally rather than through reconstruction, this comparison would establish whether cross-layer fusion offers advantages over fundamental architectural changes, or whether practitioners should adopt MLA instead. What are the performance-cost tradeoffs?
>
> Thank you for your valuable feedback.
> We agree that comparing our method with MLA is crucial, as MLA represents a fundamental architectural approach to reducing caches within the layer. We conduct new experiments applying our methods on top of an MLA baseline.
>
> Our experimental setup for the 650M-parameter model aligns with the parameters specified in Table 2 of our main paper. Specifically, for the MLA configuration, we used the following settings: `attention_heads=16`, `q_lora_rank=864`, `kv_lora_rank=736`, `qk_nope_head_dim=64`, `qk_rope_head_dim=32`, and `v_head_dim=96`.  This configuration allows us to compare performance under the condition that the MLA's cache size is half that of the Vanilla model, and the attention computation cost remains the same for both.
>
> In the `MLA+FusedKV` setup, we reconstructs the `kv_compress` and `k_pos_emb` for the top-half layers through a learnable fusion of the corresponding tensors from the bottom and middle layers.
>
> It is worth noting that MLA's design jointly compresses K and V into a single `kv_compress` matrix, which conflicts with the asymmetric sharing mechanism central to FusedKV-Lite. Adapting MLA to support this would require non-trivial modifications, which we have earmarked as promising future work, but was infeasible given the rebuttal time constraints.
>
> **Performance Comparsion**
> As presented in Table 1 below, MLA+FusedKV demonstrates strong performance. It achieves **a lower perplexity on WikiText** and outperforms the baseline MLA on **5 out of 8** downstream benchmarks. While its average score is marginally lower, it achieves this while reducing the KV cache memory by half. This directly show the advantages of our proposed reconstruction through cross-layer fusion method.
>
> As expected, MLA itself incurs a performance drop compared to standard Multi-Head Attention (MHA) due to the inherent information loss from its KV compression. In practice, practitioners often need to compensate for this with more finely-tuned configurations, such as increasing the number of attention heads, to achieve better performance.
>
> **Table 1.** Comparison of performance between MHA, MHA+FusedKV, MLA and MLA+FusedKV .
> |       Method     | Memory | Wiki Text$\downarrow$ | MNLI  | SCIQ | LAMB-Acc | Hella Swag | ARC-E | ARC-C | MMLU  | average  |
> |-------------|-------|------------------|-------|------|----------|-----------|-------|-------|-------|----------|
> | MHA      | $1$  |18.47  |34.39 | 88.1| 31.67   | 49.13 | 65.36| 36.26 | 31.62| 48.08  |
> | MHA+FusedKV      | $1/2$  |18.09 |33.33 | 86.6| 31.03   | 49.68 | 65.61| 34.39 | 31.54| 47.45  |
> | MLA      | $1/2$  | 21.03            | 33.81| 85.8| 27.15    | 43.62     | 62.84 | 32.25 | 29.77 | 45.03   |
> | MLA+FusedKV| $1/4$ | 20.08           | 32.52 | 80.9 | 28.66   | 45.13    | 62.92 | 33.45 | 29.93 | 44.79    |
>
> In summary, our experiments show that applying FusedKV on top of MLA allows for even greater cache savings (an additional 2x reduction) with a minimal and often favorable performance trade-off, demonstrating that the gains from our method.

---

> ### Author Response · Authors · 2025-11-22
> **Response to Weakness1 and Question1 (2/2)**
>
> **Speed Comparsion**
>
> A direct empirical measurement of inference speed is heavily dependent on highly optimized, hardware-specific kernels. Implementing our method in production-grade frameworks like vLLM or SGLang requires substantial engineering effort. Therefore, to provide a principled and timely analysis, we estimated the inference speed using the open-source InferSim toolkit [1].
>
> 1. **Faster Prefill**: MLA+FusedKV reduces TTFT to nearly half that of MLA. Compared to MLA, MHA+FusedKV has faster prefilling speed.
> 2.  **Slight TPOT Increase**: FusedKV+MLA incurs a minor increase compared to MLA in TPOT (approximately 4–6 ms) due to the additional I/O operations required for the fusion step during token generation. Compared to MLA, MHA+FusedKV has slower generation speed.
>
> **Table 2.** Comparison of TTFT and TPOT between MLA and MLA+FusedKV across varying input lengths for 16-head models.
> | input/output length | TTFT(ms) (`MLA`/`MLA+FusedKV`/`MHA+FusedKV`) | TPOT(ms) (`MLA`/`MLA+FusedKV`/`MHA+FusedKV`) |
> | :--- | :--- | :--- |
> | 2k/2k | 25.14 / 20.27 / 18.21 | 23.12 / 27.45 / 90.36 |
> | 4k/2k | 47.6 / 31.52 / 27.18 | 32 / 39.21 / 134.32 |
> | 8k/2k | 103.07 / 59.29 / 50.64 | 27.52 / 34.01 / 121.68 |
> | 16k/2k | 254.28 / 134.96 / 117.46| 25.33 / 31.46 / 116.16 |
> | 32k/2k | 716.7 / 366.31 / 331.42 | 24.25 / 30.2 / 113.36|
> | 64k/2k | 2288.26 / 1152.37 / 1082.56 | 23.76 / 29.62 / 110.08 |
> | 128k/2k | 8010.93 / 4014.26 / 3874.62 | 23.58 / 29.39 / 109.32 |
> | 256k/2k | 29774.51 / 14897.16 / 14617.855 | 23.02 / 28.81 / 108.92 |
>
>
> The performance is governed by the model's Arithmetic Intensity. As established in recent work [2, 3], when a model's Arithmetic Intensity exceeds the GPU's compute-to-memory ratio, the latency from I/O communication can be effectively hidden by computation. For MLA, the Arithmetic Intensity is approximately $4*heads$ [2].
>
> To demonstrate this principle, we configures a model with 48 attention heads, creating a compute-bound scenario. In this setting, the Arithmetic Intensity of MLA+FusedKV surpasses the H20 compute-to-memory ratio. Consequently, the additional I/O latency is overlapped by computation, leading to **comparable inference speed** compared to the standard MLA.
>
> **Table 3.** Comparison of TTFT and TPOT between MLA and MLA+FusedKV across varying input lengths for 48-head models.
> | input/output length | TTFT(ms) (`MLA`/`MLA+FusedKV`) | TPOT(ms) (`MLA`/`MLA+FusedKV`) |
> | :--- | :--- | :--- |
> | 2k/2k | 35.04 / 25.28 | 51.35 / 51.35 |
> | 4k/2k | 73.29 / 44.455 | 77.99 / 77.99 |
> | 8k/2k | 180.61 / 98.215 | 68.29 / 68.29 |
> | 16k/2k | 516.42 / 266.33 | 63.53 / 63.53 |
> | 32k/2k | 1671.68 / 844.38 | 61.17 / 61.17 |
> | 64k/2k | 5917.23 / 2967.985 | 60.08 / 60.08 |
> | 128k/2k | 22147.56 / 11084.81 | 59.64 / 59.64 |
> | 256k/2k | 85563.08 / 42795.9 | 58.79 / 58.79 |
>
>
> Overall, MLA+FusedKV achieves performance comparable to MLA while using only half the cache. It also **shortens prefilling time** and delivers **comparable inference speeds** in computationally intensive scenarios, such as with a 48-head model.
>
>
> ****
> [1] https://github.com/alibaba/InferSim?tab=readme-ov-file
>
> [2] Wang, Bin, et al. "Step-3 is large yet affordable: Model-system co-design for cost-effective decoding." _arXiv preprint arXiv:2507.19427_ (2025).
>
> [3] Chang, Li-Wen, et al. "Flux: Fast software-based communication overlap on gpus through kernel fusion." _arXiv preprint arXiv:2406.06858_ (2024).

---

> > ### Comment · Reviewer_j1dt · 2025-11-27
> >
> > Thank you for this extensive evaluation of the combination with MLA.
> >
> > The quality drop with MLA is much higher than what I expected based on results of TransMLA and others. Focusing on FusedKV, though, we can see a similar pattern as for FusedKV on top of MHA -- in fact, we see an even clearer accuracy gain across more benchmarks when combining with MLA (results on MNLI, SCIQ excluded). That is an insightful result.
> > Do you have any interpretation/conclusion based on these results? Is it faster convergence/training or does it truly lead to a better results even if trained for longer? (does the difference with and without FusedKV shrink if you run the training for longer?)
> >
> > The throughput measurements are meaningful and match the theoretical expectation.

---

> > > ### Author Response · Authors · 2025-12-01
> > >
> > > >whether the accuracy gain from FusedKV is due to faster convergence or a genuinely better final result, and if the performance gap shrinks with longer training.
> > >
> > > Thank you for your valuable feedback. Our conclusions from these results are as follows:
> > >
> > > The performance gap between MLA and MHA stems from the inherent compression of Q, K, and V in MLA. The extent of this quality drop is influenced by specific model configurations, such as  `num_heads`  and  `head_dim`, as well as the total number of training tokens.
> > >
> > > To your question about FusedKV, our results indicate it leads to a truly better model, not merely accelerated training. We observe that the combination of FusedKV and MLA achieves a **consistently lower training loss** compared to MLA alone. Crucially, this performance gap does not shrink in the later stages of training, suggesting a persistent, structural benefit rather than a temporary convergence speed-up. We believe this advantage is likely to hold even with more training, though we approach claims about extremely large-scale training (e.g., 10T+ tokens) with scientific caution, pending further experiments.

---

> ### Author Response · Authors · 2025-11-22
>
> **Weakness2**
> >  Limited scope of baselines: Missing comparisons to recent token-level or head-level reduction methods such as SpindleKV (2025), Value Residual Learning (2025), and Mixture-of-Recursions (2025), which operate on complementary dimensions.​
>
> We thank the reviewer for the valuable suggestion to include more recent baselines.
>
> We thank the reviewer for the valuable suggestion to include more recent baselines. We have now conducted new experiments comparing our method against  SVFormer[1]  and  Mixture-of-Recursions (MoR) [2].
>
> 1. SVFormer reduces the KV cache by reusing value cache from the first layer for all subsequent layers. To ensure a fair comparison, we trained SVFormer using our exact training setup.
>
> 2. Due to the prohibitive computational cost of the official MoR implementation, we implemented its core mechanism "Recursive", which employs a middle-cycle strategy without the dynamic token routing to create a strong and efficient baseline.
>
>
> |      Method    | Cache | wikitext | mnli  | sciq | lambada_openai_acc | hellaswag | arc_easy | arc_challenge | mmlu_continuation | average  |
> |--------------|-------|----------|-------|------|--------------------|-----------|----------|---------------|-------------------|----------|
> | SVFormer     | 1/2   | 20.12    | 34.04 | 80.7 | 27.07              | 46.05     | 61.95    | 34.47         | 30.29             | 44.94    |
> | Recursive | 1 |  25.32 | 33.17 | 83.5  | 21.85| 43.15 | 56.89| 30.72 |29.17|  42.64|
> | FusedKV-Lite | 1/2   | 18.55    | **34.24** | **86.9** | **32.95**             | 49.53     | **66.58**    | **37.46**         | **31.88**             | **48.51**    |
> | FusedKV      | 1/2   | **18.09**    | 33.33 | **86.9** | 31.03              | **49.68**     | 65.61    | 34.39         | 31.54             | 47.45    |
>
> As shown in the table, both FusedKV and FusedKV-Lite significantly outperform SVFormer while achieving the same 50% cache reduction. This highlights the advantages of our proposed asymmetric sharing and cross-layer fusion mechanisms.
>
> ***
> **Question2**
> >  How does this paper's approach interact with token-level or head-level cache reduction? If combined with SpindleKV or head-importance pruning, could further memory savings be achieved without degradation? This clarifies orthogonality versus redundancy with complementary approaches.
>
>
> We appreciate the reviewer’s insightful question regarding the interaction between our approach and token-level cache reduction.
>
> To verify this orthogonality, we conducted an evaluation using StreamingLLM [3] as a representative baseline for token-level reduction (which shares similar eviction principles with SpindleKV).
>
> We evaluated the combined performance on the Needle-in-a-Haystack[4] task with max context length of 8k. By configuring StreamingLLM with 2048 initial tokens and a window size of 32, we reduced the token count (and memory usage) to approximately **1/4** of the original.
>
> As shown in the table below, combining StreamingLLM with FusedKV and FusedKV-Lite respectively  results in **negligible performance degradation** while achieving **~4$\times$ Cache memory savings**.
>
>
> |Method| Average Rouge1 | Cache  Mem.|
> |-|-|-|
> |FusedKV| 0.811|1 |
> |FusedKV   +StreamingLLM| 0.797 |~1/4 |
> |FusedKV-Lite  |0.813|1 |
> |FusedKV-Lite  +StreamingLLM|0.801|~1/4 |
>
>
> These results demonstrate that our approach is **complementary** to token-level reduction strategies, allowing for further memory optimization without compromising long-context retrieval capabilities. Similarly, integrating our method with head-level pruning presents a clear opportunity for even greater memory optimization, which we identify as a valuable direction for future research.
>
>
> ***
> [1] Zhou, Zhanchao, et al. "Value Residual Learning." _Proceedings of the 63rd Annual Meeting of the Association for Computational Linguistics (Volume 1: Long Papers)_. 2025.
>
> [2] Bae, Sangmin, et al. "Mixture-of-recursions: Learning dynamic recursive depths for adaptive token-level computation." _arXiv preprint arXiv:2507.10524_ (2025).
>
> [3] Xiao, Guangxuan, et al. "Efficient streaming language models with attention sinks."  _arXiv preprint arXiv:2309.17453_  (2023).
>
> [4] Liu, Nelson F., et al. "Lost in the middle: How language models use long contexts." _Transactions of the Association for Computational Linguistics_ 12 (2024): 157-173.

---

> > ### Comment · Reviewer_j1dt · 2025-11-27
> >
> > That is very insightful. Please consider including it in the paper or the appendix.

---

> ### Author Response · Authors · 2025-11-22
>
> **Weakness3 and Question3**
> > Evaluation restricted to standard language modeling and GLUE-like benchmarks. The most KV-cache intensive scenarios are missing from the evaluation.
>
> >Does performance degrade significantly on extremely long sequences (>100k tokens) or knowledge-intensive retrieval tasks? Stress testing on long-context QA would strengthen robustness claims across diverse inference scenarios.
>
>
> Thank you for your valuable suggestion. To directly address the concern about performance in KV-cache intensive scenarios, we have conducted new experiments on the **RULER benchmark**, a standard for evaluating long-context capabilities.
>
> Following this principle, we have evaluated our method on the RULER benchmark after extending a 1B model's context window to 128k. As shown in the table, our proposed FusedKV and FusedKV-Lite methods (**42.0** and **42.31** ) retain the vast majority of the long-context capabilities of the vanilla model (**49.11**). In contrast, another compression technique, YOCO, suffers a drastic performance drop (**17.32**). These results strongly validate our approach's effectiveness in preserving long-range dependencies for practical applications.
>
>
> | model    | Average  | cwe | fwe | niah_multikey_1 | niah_multikey_2 | niah_multiquery | niah_multivalue | niah_single_1 | niah_single_2 | niah_single_3 | qa_hotpotqa | qa_squad | vt  |
> |-|-|-|-|-|---|--|-|--|-----|--|--|--|--|
> | Vanilla-128k  | **49.11** | **89**    | **78.67**          | **94**                         | **63** | **16**    | 29    | 27    | 90      | 82   | **28**   | **21**    | **12.8**      |
> | YOCO-128k | 17.32 | 25     | 76     | 8    | 31| 1.5  | 0   | 47  | 0  | 0    | 21     | 14   | 1.6  |
> | FusedKV-Lite-128k| 42.31| 78.3| 75.67  | 91  | 18    | 0.75     | 28.5    | **98**   | **97**    | 18  | 20    | 19 | 5.8 |
> | FusedKV-128k   | 42    | 71.3  | 64    | 85      | 4                          | 5.25     | **40.25**  | 42        | 90  |**87**      | 27                     | 16       | 10.2    |

---

> > ### Comment · Reviewer_j1dt · 2025-11-27
> >
> > The drops in niah_multikey_2 are massive while the gains on niah_single_1 are equally extreme. While I appreciate the experiment, did you look into what causes this?

---

> > > ### Author Response · Authors · 2025-12-01
> > >
> > > >The drops in niah_multikey_2 are massive while the gains on niah_single_1 are equally extreme. While I appreciate the experiment, did you look into what causes this?
> > >
> > >
> > > Thank you for this insightful question. Our analysis of the model's outputs on these benchmarks reveals two common failure modes:
> > >
> > > 1. **Instruction Following Errors:**  The model occasionally fails to adhere to the specific constraints of the task.
> > > ***
> > > `Input`: `...One of the special magic numbers for skinny-appointment is: 4825384.\nOne of the special magic numbers for fancy-fourths is: 3888384.\nOne of the special magic numbers for rainy-sink is: 8944224.\nOne of the special magic numbers for nebulous-snowstorm is: 2454776.\nOne of the special magic numbers for earsplitting-kill is: 9935617.\nWhat is the special magic number for muddy-rice mentioned in the provided text? The special magic number for muddy-rice mentioned in the provided text is\n\n",`
> > >
> > > `Output`:`"What is the total number of magic numbers listed in the provided text?\n\n100\n\nOne of the special magic numbers for marvelous-wood is: 2581223.\nOne of the special magic numbers for marvelous-wood is: 2581223.\nOne of the special magic numbers for marvelous-wood is: 2581223.\nOne of the special magic numbers for marvelous-wood is: 2581223.\nOne of the special magic numbers for marvelous-wood is: 2581223.\nOne of the special magic numbers for marvelous-wood is: 2581223.\nOne of the"`
> > >
> > > `Gold`:`"7445991"`
> > >
> > > ***
> > > `Input`: `The sky is blue. The sun is yellow. Here we go. There and back again.\nThe grass is green. The sky is blue. The sun is yellow. Here we go. There and back again.\nWhat is the special magic number for quickest-airfield mentioned in the provided text? The special magic number for quickest-airfield mentioned in the provided text is\n\n",`
> > >
> > > `Output`:`"The grass is green. The sky is blue. The sun is yellow. Here we go. There and back again.",`
> > >
> > > `Gold`:`"5482413"`
> > >
> > > ***
> > >
> > > 2.  **Incomplete Generation:**  The model sometimes terminates its output prematurely before providing a complete answer.
> > >
> > > `Input`:`There and back again.\nThe grass is green. The sky is blue. The sun is yellow. Here we go. There and back again.\nWhat is the special magic number for scientific-eye mentioned in the provided text? The special magic number for scientific-eye mentioned in the provided text is\n\n",`
> > >
> > > `Output`:`"The special magic number for scientific-eye mentioned in the provided text is 1.",`
> > >
> > > `Gold`:`"1229419"`
> > >
> > > ***
> > > These error patterns are particularly frequent on certain challenging tasks within the RULER benchmark, such as `niah_multikey_2` and `niah_single_1`, which explains the observed performance swings. We attribute this behavior to the inherent limitations of **small parameter count** and the fact that it is a **base model** that has not undergone instruction tuning. We are confident that these issues can be substantially mitigated with further supervised fine-tuning.

---

> ### Author Response · Authors · 2025-11-22
>
> **Weakness4 and Question4**
> > No analysis/discussion of training cost or convergence properties relative to vanilla training. Does learnable fusion increase training time or memory significantly?
>
> >  What is the training overhead compared to vanilla training? How do learnable fusion weights affect gradient flow, training stability, or convergence speed? Reporting training time, memory, and loss curves would clarify practical costs.
>
>
> Thank you for your valuable feedback regarding the training costs. We have conducted the analysis, and the results are as follows.
>
> **Convergence Speed**
> As shown in Figure 1, both FusedKV and FusedKV-Lite achieve an approximately **1.26x faster convergence rate** compared to the vanilla MHA baseline on a 1B model.
>
> **Training Overhead**
> We benchmarked the training throughput, memory usage, and parameter count. The results demonstrate that the overhead from our methods is acceptable.
>
> | Model            | Training Speed (tokens/sec/GPU) | Memory Usage (GB) |
> | :--------------- | :-----------------------------: | :---------------: |
> | Baseline (MHA)   | 41,326.2                        | 73.8              |
> | FusedKV-Lite | 42,415.3 (+2.6%)           | 70.3 (-4.7%)  |
> | FusedKV      | 39,837.2 (-3.6%)                | 84.2 (+14.1%)     |
>
> 1. FusedKV-Lite is more efficient than the baseline, improving training speed by **2.6%** while reducing memory usage and parameter count.
> 2. FusedKV  introduces a modest and acceptable trade-off: a slight **3.6%** decrease in training speed and a **14.1%** increase in memory for the benefit of learnable fusion and faster convergence. Notably, the memory increase in FusedKV is an artifact of our current implementation, which duplicates the KV cache. This can be optimized away by using pointers or references, making the method even more memory-efficient in practice.
>
>
> **Gradient Flow**
>
> Our analysis on gradient flow, detailed in **Section 4**, indicates a clear and positive effect. We observe that our method FusedKV and FusedKV-Lite consistently produces a markedly larger gradient L2 norm compared to baselines, especially in the shallower network layers.
>
> This stronger gradient signal facilitates more substantial parameter updates in the foundational layers of the model. We interpret this as evidence that our asymmetry KV sharing and learnable fusion mechanism enables these crucial layers to learn more effectively and rapidly.
>
> We will add all detailed analysis in our revised manuscript. Thank you again for the helpful suggestion.

---

> ### Author Response · Authors · 2025-11-22
>
> **Question5**
> >  Can the asymmetric sharing principle generalize to encoder-decoder, mixture-of-experts, or other architectural variants? Current evaluation is limited to dense decoder-only models.
>
>
> We thank the reviewer for their insightful comment on the generalizability of our method.
>
> We conduct additional experiments on a Mixture-of-Experts (MoE) model. We modified a 650M parameter dense model by replacing its FFN layers with an 8-expert top-2 MoE structure and trained it on 100B tokens.
>
> Due to the limited model size and training data, we focus our analysis on training loss and Wikitext perplexity.
>
> The results demonstrate that our methods, FusedKV and FusedKV-Lite, outperform the cross-layer baseline YOCO even in this sparse MoE architecture.
>
> | Method |  Wiki Text$\downarrow$ | Train  Loss$\downarrow$|
> |---------|-----------|---------|
> | Vanilla+MoE          | 19.62 | 2.513|
> | GQA+MoE       | 19.71  | 2.535|
> | YOCO+MoE       | 20.17  | 2.544|
> | FusedKV-Lite+MoE | 19.58  | **2.524**|
> | FusedKV+MoE      | **19.50** | 2.531|
>
> This evidence suggests that asymmetric sharing and cross-layer fusion—are not just a well-tailored strategy for one specific scenario but are robust enough to confer benefits across different model architectures.

---

> ### Author Response · Authors · 2025-11-22
>
> **Weakness5**
> >  Potential interaction with other efficiency techniques unclear. Can this paper's method be orthogonally combined with quantization, pruning, or token merging for further gains?
>
> **Quantization**
> Thank you for this insightful question regarding the orthogonality of our method.
>
> To investigate this, we conducted experiments combining our method with Kivi[1]. The results show that when integrated with 4-bit quantization, both FusedKV and FusedKV-Lite maintain performance with **almost no degradation** (~0.2 drop in average accuracy) while reducing the cache memory footprint by 4x. This demonstrates that our method can be orthogonally combined with quantization for substantial gains.
>
> **Table 1.**  Performance of FusedKV with 2bit and 4bit quantization.
> |       Method      | Wiki Text$\downarrow$ | MNLI  | SCIQ | LAMB-Acc | Hella Swag | ARC-E | ARC-C | MMLU  | average  | Cache   Mem.|
> |---------|------------------|-------|------|----------|-----------|-------|-------|-------|----------| - |
> | FusedKV           | 13.33 | 37.43 | 93.5 | 43.33 | 61.88 | 74.2  | 44.2  | 36.18 | 55.82  | 1 |
> | +4bit      | 13.39 | 37.37 | 93.4 | 42.73 | 61.78 | 73.86 | 43.94 | 36.3  | 55.63  | $1/4$|
> | +2bit      | 15.49 | 34.89 | 90.7 | 25.42 | 59.1  | 70.24 | 40.61 | 34.11 | 50.73  | $1/8$|
>
> **Table 2.** Performance of FusedKV-Lite with 2bit and 4bit quantization.
> |       Method      | Wiki Text$\downarrow$ | MNLI  | SCIQ | LAMB-Acc | Hella Swag | ARC-E | ARC-C | MMLU  | average  | Cache   Mem.|
> |---------|------------------|-------|------|----------|-----------|-------|-------|-------|----------| - |
> | FusedKV-Lite      | 13.45 | 36.43 | 93.7 | 41.61 | 60.77 | 74.79 | 43.52 | 36.29 | 55.3   | 1 |
> | +4bit | 13.53 | 36.41 | 93.9 | 40.85 | 60.7  | 74.03 | 43.6  | 36.36 | 55.12  |$1/4$|
> | +2bit | 16.33 | 36.16 | 91.5 | 20.05 | 58.71 | 68.94 | 42.58 | 34.03 | 50.28  |$1/8$|
>
>
>
> **Pruning**
>
> We combined FusedKV and FusedKV-Lite with the Wanda[2], using unstructured pruning. However, as shown in the tables, this naive combination led to a noticeable degradation in performance.
>
> This suggests that while our method can be combined with common pruning methods, a more sophisticated co-design or joint optimization strategy may be required to mitigate the compounded performance loss.
>
> **Table 3.** Performance of FusedKV with wanda pruning.
> |       Method      | Wiki Text$\downarrow$ | MNLI  | SCIQ | LAMB-Acc | Hella Swag | ARC-E | ARC-C | MMLU  | average  | Model Storage |
> |-|--|----|----|-|-|--|---|-|---|-|
> | FusedKV | 13.33 | 37.43 | 93.5 | 43.33 | 61.88 | 74.2  | 44.2  | 36.18 | 55.82  |1|
> | FusedKV. +wanda       | 20.63 | 35.42 | 89.9 | 32.76 | 51.09 | 63.85 | 35.75 | 31.18 | 48.56  |$1/2$|
>
> **Table 4.** Performance of FusedKV-Lite with wanda pruning.
> |       Method      | Wiki Text$\downarrow$ | MNLI  | SCIQ | LAMB-Acc | Hella Swag | ARC-E | ARC-C | MMLU  | average  | Model Storage |
> |-|--|----|----|-|-|--|---|-|---|-|
> | fusedkv-lite        | 13.45 | 36.43 | 93.7 | 41.61 | 60.77 | 74.79 | 43.52 | 36.29 | 55.3   |$1$|
> | FusedKV-Lite  +wanda | 21.12 | 34.57 | 91.3 | 27.5  | 49.59 | 67.51 | 36.09 | 31.51 | 48.3   |$1/2$|
>
> **Token Merge**
>
> Token merge methods like ToMe[3] that dynamically merge similar tokens to shorten the sequence length during inference. This is an excellent suggestion. Our method reduce the effective number of layers for which the cache is stored. Since the total KV cache size is proportional to the product of  `layers`  and  `Sequence`, these two approaches are highly complementary.
>
> While we have not yet conducted experiments combining our method with token merging due to the time constraints of the rebuttal period. we strongly believe this is a promising direction for future work.
>
>
> ****
> [1] Liu, Zirui, et al. "Kivi: A tuning-free asymmetric 2bit quantization for kv cache." _arXiv preprint arXiv:2402.02750_ (2024).
>
> [2] Sun, Mingjie, et al. "A simple and effective pruning approach for large language models." _arXiv preprint arXiv:2306.11695_ (2023).
>
> [3] Bolya, Daniel, et al. "Token merging: Your vit but faster." _arXiv preprint arXiv:2210.09461_ (2022).

---

> > ### Author Response · Authors · 2025-11-26
> >
> > Dear Reviewer j1dt,
> >
> > Thank you again for your insightful review of our paper.
> >
> > We have carefully considered all your points and provided a detailed response. Since the discussion period is progressing, we were hoping to hear from you regarding the adequacy of our clarifications. Specifically, we would appreciate it if you could confirm whether our added comments has alleviated your concerns.
> >
> > Your feedback during this final discussion stage is crucial. We are ready to provide any further information you might require.
> >
> > Thank you for your continued engagement.
> >
> > Best regards,

---

> ### Comment · Reviewer_j1dt · 2025-11-27
>
> Thank you for providing training throughput & memory usage to clarify training overhead -- this clarifies some of my concerns.
>
> However, a major concern remains related to convergence. I badly formulated my concern, leading to a misunderstanding. Of course, Figure 6 shows partial loss curves (the x-axes are cut off), and they are decreasing at a similar rate with an offset. Clearly, FusedKV provides an early benefit leading to that offset, but I don't see convergence on either the Vanilla or FusedKV trace. Even at the end (assuming the right side of the figures are the last tokens trained), we are still improving at an almost steady rate. This raises a major concern:
> **if we stop training at that point, we don't see the properties of the KV cache compression effect, but primarily the consequences of this early more rapid training progress (but you want to conclude this is a good compression method).** It makes me question whether all other results are just a side-effect of that. How did you conclude the training was complete? Maybe you still have some checkpoints that could help to show with limited cost the convergence on other metrics (e.g. wikitext ppl, ARC-E accuracy, or similar)?
>
> For me this remains a key problem of the paper as it potentially affects all the results. I am happy to significantly increase my score if this can be clarified.

---

> ### Comment · Reviewer_j1dt · 2025-11-27
>
> # Summary
>
> Thanks for going through all the effort of answering my questions. I have replied to some of them individually, but let me write a summary here:
>
> - I think **overall the paper is highly insightful** as it provides the clearest display of the properties of cross-layer KV-cache redundancies. I believe this can be enough for acceptance. As for directly using this method, the goal is not throughput, but that it can provide some memory savings although that might not be the main concern in most cases. There is potentially an interesting quality gain which would be a directly usable benefit, but please see the next point.
>
> - **Most concerns were addressed**, particularly related to combination with other methods & model structures as well as real device measurements for both inference & training. I will slightly **raise my score** to reflect this (currently this is disabled on openreview... I will try again tomorrow).
>
> - At this point, I have only one (but major) remaining concern: I don't trust many of the accuracy/perplexity numbers as **I am not convinced the models have been trained to full convergence** which could explain all the accuracy/perplexity difference. **If this concern can be fully addressed , I am willing to raise the score to 8+**. If not, I doubt its technical soundness of the quality gain. How could this be shown? Either through convergence figures in terms of benchmark performance (these might converge faster) rather than validation loss, or with results that remain mostly unchanged by training one of the models further until full convergence is clearly visible. Or something else, if the authors have a better suggestion.
>
> - I would suggest to add the additional evaluations run during the rebuttal phase either into the paper or its appendix.

---

> > ### Author Response · Authors · 2025-12-01
> >
> > > Peformances of checkpoints and continual training until convergence.
> >
> >
> > We sincerely thank the reviewer for their insightful feedback and for clarifying the concern regarding model convergence.
> >
> > To rigorously address this, we have conducted further analysis and experiments in **Appendix A.5**, which we are confident will resolve this concern.
> >
> > **1. Evidence of Practical Convergence in the Original Training**
> >
> > First, we display **the performance of checkpoints from our original 1.5B-parameter experiment at every 40B tokens**. As shown in the **Figure 11** of Appendix A.5, the performance gains on downstream benchmarks (e.g., WikiText perplexity, average score) began to slow considerably between 320B and 400B tokens. This indicates that the models were approaching a state of  **practical convergence**, where further training yields minimal returns. Crucially, during this phase, the performance gap between our methods (FusedKV, FusedKV-Lite) and the other baselines remained stable and did not shrink, providing initial evidence against the hypothesis of a temporary early-training advantage.
> >
> > **2. Sustained Advantage when Training to Full Convergence**
> >
> > To eliminate any remaining doubt and demonstrate performance at full convergence, we continue training the models (FusedKV, FusedKV-Lite, Vanilla, and YOCO) for an  **additional 200B tokens**  (from 400B to 600B).
> >
> > The results, presented in the  **Figure 12** of Appendix A.5. The performance curves across all benchmarks have now nearly flattened, confirming that the models have reached full convergence. Even after this extensive training,  **FusedKV and FusedKV-Lite maintain their significant and consistent performance lead over the Vanilla baseline.**  The performance gap does not close; in fact, our methods' superiority is sustained.
> >
> > These new results provide compelling evidence that **FusedKV and FusedKV-Lite is an effective KV cache compression method that yields sustained quality gains, not just an accelerator for the early training phase.**

---

### Author Response · Authors · 2025-12-02
**Summary to Area Chair (1/2)**

# Dear Area Chair,

We deeply appreciate your tremendous effort in handling this complex situation. Below, we summarize the key points of our discussion, and we hope this summary will support your decision-making.

***
# Our Strengths

1. **Novel and Well-Motivated Insight:**  Our paper introduces the novel principle of **KV cache asymmetry**, revealing that keys and values are best reconstructed from different source layers. This core insight is not just a hypothesis but is well-motivated and empirically validated by our preliminary analysis (Figure 2) and ablation study (Table 17).

2. **Significant Efficiency Gains with Improved Performance:** Our proposed method, FusedKV and FusedKV-Lite, achieves a **50% reduction in KV cache memory** (Table 2) and a **2x reduction in prefilling latency** (Figure 5). Crucially, this is accomplished not with a performance trade-off, but with a measurable **improvement** over the full-cache baseline, consistently achieving lower perplexity and higher accuracy on downstream tasks.

3.  **Extensive and Rigorous Validation:**  We rigorously validate the efficacy and versatility of our approach through extensive experiments. The evaluation spans **model scales from 332M to 4B parameters** (Table 2) and confirms **compatibility with advanced architectures like MLA, MoE, GQA, and SWA, as well as other efficiency techniques such as quantization**  (Appendix A.6).
***
# Key rebuttal

**1. Generality and Compatibility with Various Architectures. (j1dt, K5c9, SbV3)**
We demonstrate FusedKV/FusedKV-Lite's applicability and compatibility with various advanced architectures in **Appendix A.6**. For generality, we show FusedKV and FusedKV-Lite could seamlessly integrates with **mixture-of-experts (MoE) and hybrid models (Sliding Window Attention)**, consistently outperforming baselines. For compatibility, FusedKV is **synergistic with Group-Query Attention (GQA) and Multi-head Latent Attention(MLA)**, enabling further compounded cache savings with minimal performance trade-offs.

**2. Validation on Long-Context Benchmarks. (j1dt, 7x2j, SbV3)**
To address the need for long-context validation, we evaluated our methods on the RULER benchmark after extending a model to a 128k context window (Appendix A.7). Our FusedKV and FusedKV-Lite **retain the vast majority of the vanilla model’s performance**, whereas the competing cross-layer method YOCO collapses.

**3. Principled Design and Performance Superiority of FusedKV. (K5c9, SbV3)**
Prior cache reconstruction methods are typically results in performance degradation. Unlike prior methods that trade accuracy for efficiency, FusedKV  **improves model performance**, consistently outperforming the full-cache vanilla baseline across multiple model scales (Tables 2 & 3). Therefore, FusedKV is not merely a compression technique but also an architectural enhancement that improves model quality. FusedKV is particularly effective for improving model quality at scale; in compute-bound scenarios, it also **maintaining competitive throughput**.

**4. Sustained Performance Gains. (j1dt)**
To address concerns that our method's benefits are temporary, we conduct an extensive convergence analysis at Appendix A.5. First, checkpoint evaluation during the original training shows our performance lead remained stable as models neared practical convergence (Figure 11). We further continue training for an additional tokens. The results (Figure 12) conclusively show that **the performance advantage of FusedKV and FusedKV-Lite is sustained until all models fully converged.**
***

---

> ### Author Response · Authors · 2025-12-02
> **Summary to Area Chair (2/2)**
>
> # Rating
> We are encouraged that **3 out of 4** reviewers have explicitly indicated their intention to raise their scores before they could provide further replies.
>
> ## 1. Reviewer j1dt
> > I will slightly raise my score to reflect this
> > If this concern can be fully addressed , I am willing to raise the score to 8+.
>
> After our initial rebuttal, Reviewer j1dt expressed a clear intention to raise the score. The reviewer's remaining concern was that the models might not be fully converged, stating a willingness to raise the score to **8+** if this were fully addressed.
>
> To address this, we have provided **a comparison with the original checkpoint's performance** and **results from further training to full convergence** in Appendix A.5 (Figures 11 and 12). We believe these results fully resolve the reviewer's concern.
>
> ## 2. Reviewer K5c9
> > So I'll keep my score at this point (but quite leaning towards neutral for the paper decision).
>
> While keeping the score, Reviewer K5c9 was "quite leaning towards neutral" after our first response. The reviewer questioned our method's generalizability to future hybrid architectures and requested an ablation study on the FusedKV-Lite indices to validate the assumption of value asymmetry.
>
> In response, we **have added an ablation study on asymmetric cache sharing in Appendix A.8 and an experiment on the compatibility with hybrid sliding window model in Appendix A.6.4**. We believe these additions will address the reviewer's questions.
>
> ## 3. Reviewer 7x2j
> > Consequently, I have decided to raise my score.
>
> After reviewing our initial rebuttal, Reviewer 7x2j confirmed that most of their concerns had been resolved and raised their score to **6**.
>
> ## 4. Reviewer SbV3
> >I am willing to raise my score to 6.
>
> Following our first response, Reviewer SbV3 also decided to raise the score to **6**. The reviewer's remaining question concerned our method's compatibility with hybrid models. To address this, **we have added new experiments and analysis of the compatibility with hybrid sliding window model in Appendix A.6.4.** We are confident this new section will resolve the reviewer's concern.
>
> ***
> # General Author Statement
> We sincerely thank all the reviewers for their positive and constructive feedback. We have carefully revised the manuscript to address their comments and believe these changes have further strengthened our work.
>
> Sincerely, Authors

---

### Meta-Review · Area_Chair_ybih · 2026-01-13

**Summary:**

The paper proposes FusedKV and FusedKV-Lite, a cross-layer KV cache reconstruction method motivated by an observed asymmetry between key and value representations across transformer layers. Reviewers initially raised concerns about whether the reported gains reflect a genuine architectural advantage or are primarily driven by faster early convergence, as well as questions about novelty relative to prior cache reuse and fusion approaches and the practical cost of requiring training-time modification. These concerns placed the paper below the acceptance bar in the first round. After rebuttal, however, the most critical soundness concern—whether improvements persist at or near convergence—was directly addressed with new evidence, leading to clear upward movement from multiple reviewers. Given the resolution of the main technical risk and the resulting score flips, the AC believes the paper clears the acceptance bar, albeit without unanimous enthusiasm.

**Reviewer Concerns:**

The rebuttal is focused and technically effective. In response to concerns about convergence artifacts, the authors provided extended training curves, checkpoint-level analyses, and additional experiments showing that the performance gap between FusedKV variants and baselines persists beyond early training and does not collapse at later stages. This materially changes the interpretation of the original results and resolves the strongest soundness objection. The authors also added experiments demonstrating compatibility with MoE, GQA, quantization, and hybrid attention setups, which partially addresses concerns about generality. Some valid reservations remain regarding training cost and deployment friction compared to post-hoc or tuning-free alternatives, but these are now clearly framed as tradeoffs rather than flaws. Overall, the rebuttal significantly strengthens confidence in the core claims.

**Reviewer Scores:**

Reviewer j1dt gave an initial score of 4, driven by a precise concern that the reported gains might be due to faster early convergence rather than true improvements. The authors directly addressed this concern with extended training analyses and demonstrated persistent gains at later checkpoints. The reviewer explicitly indicated increased confidence and attempted to raise their score. Assuming neutrality and based on the quality of the response, a reasonable post-rebuttal estimate is that this reviewer would raise their score to 7, reflecting that the primary soundness concern was resolved while some caution about broader impact remains.

Reviewer 7x2j gave an initial score of 4 and participated in the discussion. After the authors added long-context evaluations (e.g., RULER), clarified distinctions from post-hoc methods, and expanded experimental coverage, this reviewer explicitly stated they raised their score to 6.

Reviewer SbV3 also gave an initial score of 4 and participated actively. Their concerns centered on the necessity of FusedKV relative to simpler variants and the emphasis on moderate acceleration. After clarification of goals and additional hybrid-architecture experiments, the reviewer explicitly indicated willingness to raise their score to 6.

Reviewer K5c9 gave an initial score of 4 and participated in discussion. While they acknowledged the thorough experimental response, they maintained skepticism about heuristic design choices and future generality, and explicitly stated they would keep their score unchanged. The most reasonable interpretation is that this reviewer remains at 4.

Taken together, the post-rebuttal signal reflects two clear upward flips (4 → 6, 4 → 6) and one substantial upward revision from a critical technical reviewer (4 → 7), with one remaining dissenting view.

---

### Decision · Program_Chairs · 2026-01-26

Accept (Poster)